# Biosynthesis of GMGT lipids by a radical SAM enzyme associated with anaerobic archaea and oxygen-deficient environments

Yanan Li[1,2], Ting Yu[3], Xi Feng[1], Bo Zhao[1], Huahui Chen[1], Huan Yang[4], Xing Chen[5], Xiao-Hua Zhang[5], Hayden R. Anderson[6], Noah Z. Burns[6], Fuxing Zeng[3] ✉, Lizhi Tao[2] ✉ & Zhirui Zeng[1] ✉

Archaea possess characteristic membrane-spanning lipids that are thought to contribute to the adaptation to extreme environments. However, the biosynthesis of these lipids is poorly understood. Here, we identify a radical *S*-adenosyl-L-methionine (SAM) enzyme that synthesizes glycerol monoalkyl glycerol tetraethers (GMGTs). The enzyme, which we name GMGT synthase (Gms), catalyzes the formation of a $C(sp^3)$–$C(sp^3)$ linkage between the two isoprenoid chains of glycerol dialkyl glycerol tetraethers (GDGTs). This conclusion is supported by heterologous expression of gene *gms* from a GMGT-producing species in a methanogen, as well as demonstration of in vitro activity using purified Gms enzyme. Additionally, we show that genes encoding putative Gms homologs are present in obligate anaerobic archaea and in metagenomes obtained from oxygen-deficient environments, and appear to be absent in metagenomes from oxic settings.

Archaeal GDGTs (glycerol dialkyl glycerol tetraethers) membrane-spanning lipids provide structural advantages that help archaea adapt to extreme environments[1] and serve as crucial as lipid biomarkers in the reconstruction of paleoclimate changes[2]. Recent advancements in unraveling GDGTs biosynthetic pathways have laid a strong biological foundation for the effective utilization of GDGTs proxies. The key enzymes responsible for GDGTs biosynthesis, particularly the processes of $C(sp^3)$-$C(sp^3)$ cross-linking[3,4] and cyclization[5], belong to the radical *S*-adenosyl-L-methionine (SAM) superfamily. These enzymes harbor a radical SAM [4Fe–4S] cluster that utilizes SAM to generate a highly reactive 5′-deoxyadenosine radical (5′-dAdo•) via the reductive cleavage of SAM. The 5′-dAdo• then initiates diverse radical chemistries through either hydrogen atom (H-atom) abstraction or substrate adenosylation[6]. In the biosynthesis of GDGTs, the orchestrated utilization of such a high-

energy radical proves highly advantageous for forming a challenging $C(sp^3)$–$C(sp^3)$ linkage.

GMGTs (glycerol monoalkyl glycerol tetraethers) constitute an important subset of GDGTs derivatives, distinguished by an additional C–C linkage between the two isoprenoid chains[7]. These unique lipids have been identified in multiple thermophilic archaea and are thought to contribute to increased membrane rigidity at high temperature[8]. This is consistent with the observed positive correlation between GMGTs abundance and temperature in global peat environments[9]. However, findings from culture experiments exhibited a conflicting trend, wherein the abundance of GMGTs decreased as temperature increased in the archaeon *Pyrococcus furiosus*[10]. This result suggests GMGTs formation was not solely in response to environmental heat stress. Nevertheless, the lack of knowledge regarding their biosynthesis, biochemical mechanisms, and comprehensive biological sources

[1]Department of Ocean Science and Engineering, Southern University of Science and Technology, Shenzhen, China. [2]Department of Chemistry, Southern University of Science and Technology, Shenzhen, China. [3]Department of Systems Biology and Institute for Biological Electron Microscopy, Southern University of Science and Technology, Shenzhen, China. [4]State Key Laboratory of Biogeology and Environmental Geology, China University of Geosciences, Wuhan, China. [5]Frontiers Science Center for Deep Ocean Multispheres and Earth System, College of Marine Life Sciences, Ocean University of China, Qingdao, China. [6]Department of Chemistry, Stanford University, Stanford, USA. ✉e-mail: zengfx@sustech.edu.cn; taolz@sustech.edu.cn; zengzr@sustech.edu.cn

has hindered our understanding of their physiological significance. In this study, through the integration of in vivo gene expression and in vitro enzymatic activity assays, we have identified the key gene responsible for GMGTs biosynthesis. Moreover, this gene is exclusively present in anaerobic archaeal genomes and metagenomes obtained from oxygen-deficient environments, suggesting the potential association of GMGTs producers with oxygen-deficient habitats.

## Results

### Discovery and identification of GMGTs synthase in vivo and in vitro

The biosynthesis of GMGTs likely occurs via direct generation of a C−C bridge between two isoprenoid chains on GDGTs, i.e., forming a covalent bond between two methyl groups[11]. Inspired by the radical SAM (RS) enzyme Tes[3] and its homolog MJ0619[4] that activate inert sp³-hybridized carbon centers, we searched the genome for candidate radical SAM enzymes. It has been observed that *Thermococcus guaymasensis* produces GMGTs[12] while its close sister species *Thermococcus kodakarensis* does not[13]. Similarly, in the case of *Pyrococcus furiosus* and *Pyrococcus yayanosii*, the former produces GMGTs[10], whereas the latter does not[13]. By comparing the homology of all radical SAM enzymes (pfam04055) between species with or without GMGTs formation, we successfully found a promising candidate protein (WP_062369926.1 from *T. guaymasensis*, or AAL80771.1 from *P. furiosus*), which is present in GMGTs-forming species but absent in species lacking GMGTs formation (Supplementary Table 1). These proteins containing a canonical $CX_3CX_2C$ motif which provides three cysteine (Cys) residues to ligate a RS [4Fe–4S] cluster for performing radical SAM chemistry (Supplementary Fig. 1). We thought it highly likely that this protein is responsible for forming the bridging covalent bond within isoprenoidal GDGTs that leads to the formation of GMGTs.

To test our hypothesis, we conducted in vivo studies using an engineered methanogen *Methanococcus maripaludis* as the host for heterologous expression. This particular methanogen harbors the *tes* gene[3] and can produce GDGT-0 as the substrate for GMGTs synthase (Fig. 1a). We also cloned and expressed the gene homolog of *WP_062369926.1* from the methanogen *Methanothermococcus*

*okinawensis* which produces GMGTs[14], known as *METOK_RSO4425*, to improve the expression efficiency of exogenous protein in the methanogen host (Fig. 1b left). The clones were cultured at 37 °C for 7 days and then harvested for lipid analysis with reverse-phase ultra-high performance liquid chromatography mass spectrometry (UPLC-MS/MS). The results clearly showed the production of a lipid compound with $m/z = 1300.3042$, two Daltons less than the $m/z$ of GDGT-0 (1302.3259). In addition, the fragmentation pattern from a single isoprenoid chain in tandem MS/MS of GDGT-0 is absent in the pattern of the product (Supplementary Fig. 2). This is consistent with published spectra[7] and supports the formation of a C−C bond between the two isoprenoid chains. This analysis confirms the in vivo formation of GMGT-0, indicating that the radical SAM enzyme encoded by *METOK_RSO4425* is responsible for the biosynthesis of GMGTs (Fig. 1a). We have thus named this enzyme GMGTs synthase (Gms).

To confirm the function of Gms, we purified Gms enzyme and performed in vitro studies. Recombinant Gms from *M. okinawensis* was heterologously expressed in *Escherichia coli* BL21 (DE3) and purified anaerobically using *Strep*-Tactin affinity chromatography. A mutant strain of *Sulfolobus acidocaldarius*, which produces GDGT-0 as its dominant membrane lipid[5], was employed for isolating the substrate GDGT-0 using preparative LC-MS. An in vitro activity assay was then performed by incubating Gms anaerobically with 30 equiv. of reductant dithionite (DTH), 30 equiv. of SAM, and 10 equiv. of GDGT-0 at 37 °C for 72 h. The presence of GMGT-0 was confirmed, with a yield of approximately 4.8% (Fig. 1b right, Supplementary Figs. 3, 4). These results unequivocally demonstrate that Gms catalyzes the formation of a C−C linkage between two isoprenoid chains on GDGT-0 to produce GMGT-0 (Fig. 1a).

To further elucidate the details of GMGT-0 biosynthesis, we employed electron paramagnetic resonance (EPR) spectroscopy to characterize the Fe-S cluster(s) in purified Gms. EPR spectroscopy is a technique that probes paramagnetic species, e.g., organic radicals, metal ions with specific oxidation states, or multi-nuclear clusters in certain redox states[15]. As shown in Fig. 2, the EPR spectrum of the as-purified Gms reduced by DTH reveals a complex Fe-S cluster signal. Subsequent addition of SAM results in an immediate spectral change, suggesting SAM binds to the radical SAM [4Fe–4S]⁺ cluster and

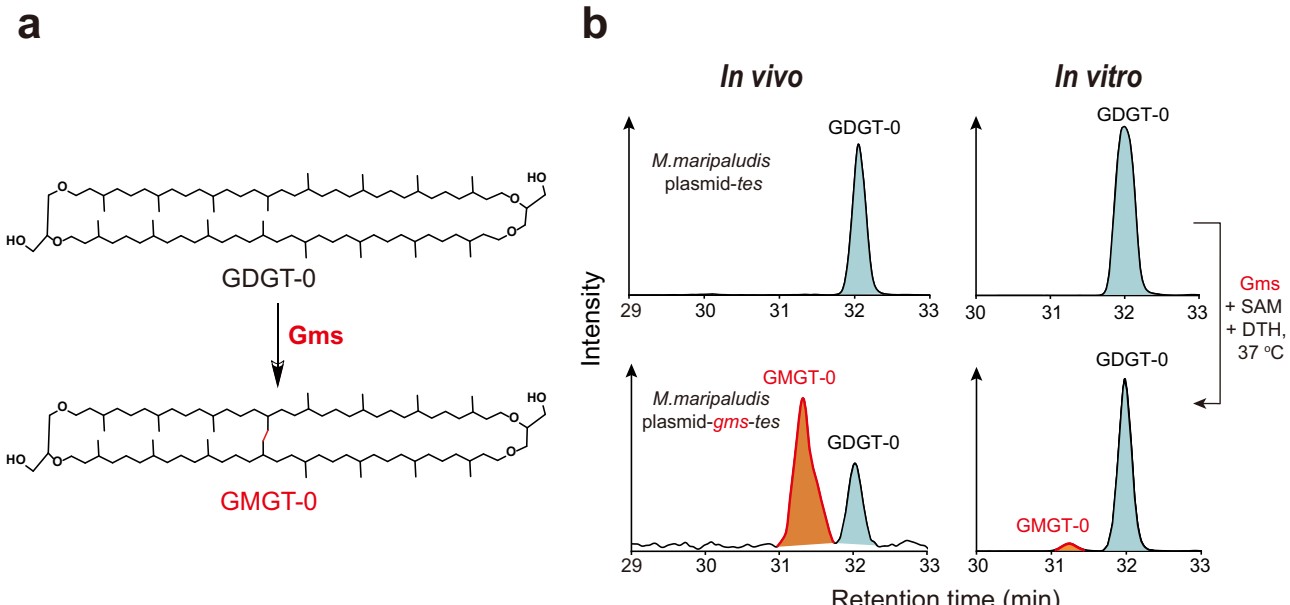

**Fig. 1 | Identification of GMGTs synthase by in vivo and in vitro assays. a** GMGT-0 biosynthesis pathway. **b** UPLC-MS extracted ion chromatograms of lipid extracts from *M. maripaludis* with plasmid pMEV4-*tes* showing production of GDGT-0, and with plasmid pMEV4-*gms-tes* showing production of GMGT-0. The in vitro biochemical reaction reveals the synthesis of GMGT-0 via incubation of the purified Gms protein with GDGT-0, SAM and DTH at 37 °C for 72 h.

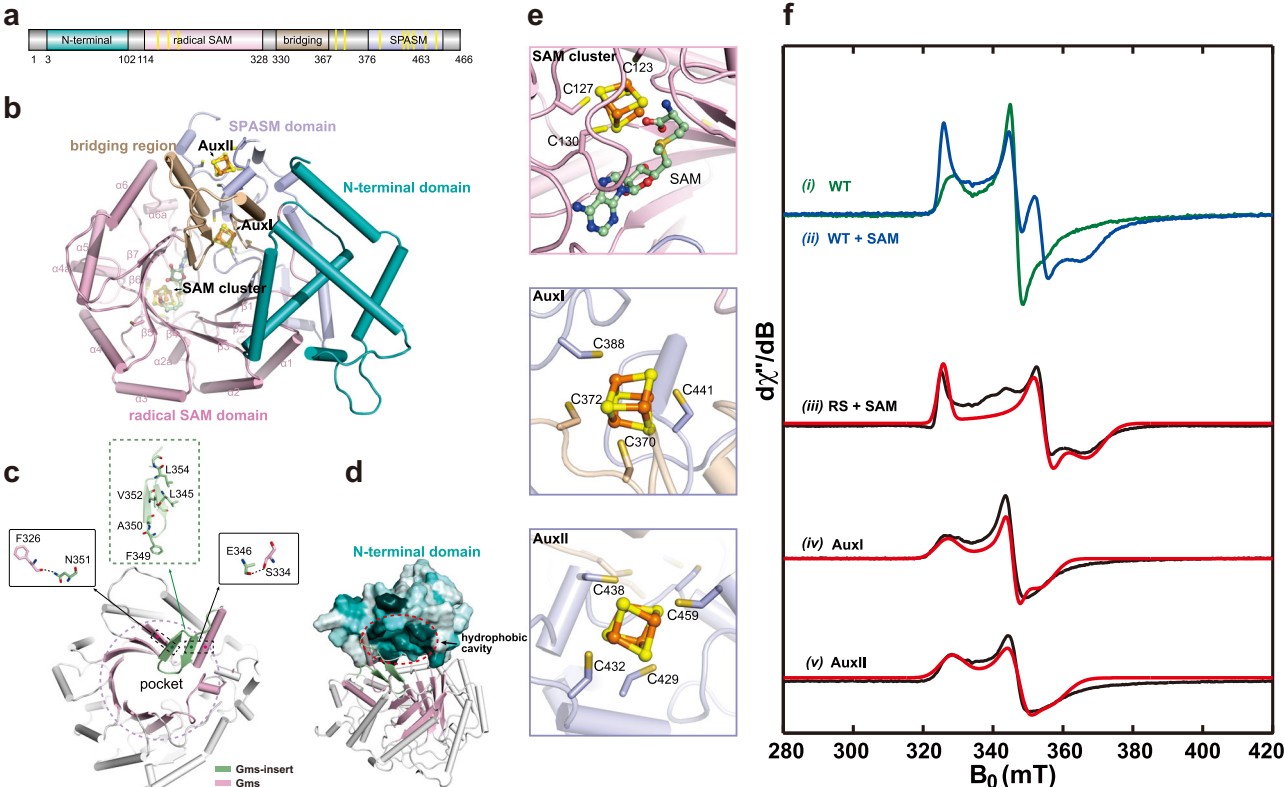

**Fig. 2 | EPR spectroscopic characterization and AlphaFold2 predicted structure model of Gms. a, b** Domain organization (**a**) and the overall architecture of Gms (**b**) showing an N-terminal domain comprised of six helices, a conserved RS domain, a bridging region with two helices, a β-harpin motif, and a C-terminal SPASM domain. **c** The interaction between the β-hairpin and the pocket involves two hydrogen bonds between F326 and N351, as well as between S334 and E346. **d** Cavity map for the active site of Gms reveals the hydrophobic pocket. **e** Representations of the three [4Fe–4S] clusters in the structure model of Gms. Protein is shown as a cartoon, SAM, and [4Fe–4S] clusters are shown in ball-and-stick models. **f** X-band CW EPR spectra of various Gms samples (i) the wild-type (WT) Gms reduced by DTH; (ii) subsequently adding SAM to DTH-reduced Gms; (iii) the mutant sample of Gms (C438AC370AC372A) containing only radical SAM (RS) cluster reduced by DTH, and incubated with SAM; (iv) the mutant sample of Gms (C438AC123AC127A) containing only AuxI cluster reduced by DTH; (v) the mutant sample of Gms (C123AC127A C370AC372A) containing only AuxII cluster reduced by DTH. Black, blue and green traces are experimental spectra, and the red traces are simulated spectra.

significantly affects the electronic structure of the cluster. Additionally, we observed more than one Fe−S cluster signal in the spectra. In order to deconvolute the signal, we employed AlphaFold2[16] to predict the structure of Gms. The model shows that Gms contains four domains, including an N-terminal domain comprised of six helices, a conserved radical SAM domain (RS-domain), a bridging region with two helices and a β-hairpin motif, and a C-terminal SPASM domain (Fig. 2a and Supplementary Fig. 1). By aligning with the atomic structure of SPASM-domain containing radical SAM enzyme SuiB (PDB: 5V1T)[17], three [4Fe−4S] clusters (RS, AuxI and AuxII clusters) were displayed in our Gms model, consistent with the complex Fe−S cluster signal observed in the EPR spectra (Fig. 2b and Supplementary Fig. 1). With the aid of this structural model, we prepared corresponding Gms mutants by knocking out two [4Fe−4S] clusters, leaving only one cluster in each (Fig. 2e, f). The **g** tensors for the three clusters were clearly determined via spectral simulation, i.e., [2.051, 1.884, 1.817] for the SAM-bound radical SAM cluster, [2.045, 1.930, 1.885] for the AuxI cluster, and [2.036, 1.924, 1.871] for the AuxII cluster (Fig. 2f). The **g** tensors are characteristic of two auxiliary [4Fe−4S] clusters, but the g-tensor for SAM-bound radical SAM cluster is distinct from the typical tensor with $g_1 \sim 2.001$[6,18]. In addition, we did not observe a SAM response for both AuxI and AuxII clusters (Supplementary Fig. 5), suggesting that SAM does not bind to the auxiliary clusters as it does for the radical SAM cluster. This further corroborates the structural model showing the two auxiliary clusters fully ligated by four Cys residues.

The radical SAM domain of Gms encompasses three-quarters of the triosephosphate isomerase (TIM) barrel fold and features a site-differentiated [4Fe−4S] cluster, which is shared among all RS enzymes (Fig. 2b). Gms exhibits a substrate-binding pocket similar to Tes, but this pocket is significantly reduced in size due to the insertion of a β-hairpin within the bridging region, rendering it incapable of accommodating a GDGT-0 substrate adequately (Supplementary Fig. 6). The interaction between the β-hairpin and the pocket primarily involves two hydrogen bonds between F326 and N351, as well as S334 and E346 (Fig. 2c). On the opposite side of the β-hairpin, there are multiple hydrophobic amino acids. Therefore, it is hypothesized that during GDGT-0 substrate binding, this hairpin flips outwards, providing sufficient space for substrate binding (Supplementary Fig. 7). Additionally, the hydrophobic cavity from the N-terminal domain of Gms, which faces the substrate-binding pocket (Fig. 2d and Supplementary Fig. 7), may also be involved in substrate binding (Fig. 2d and Supplementary Fig. 7).

We then compared the structural models of four RS enzymes, including Gms, Tes, GrsA and SuiB (PDB: 5V1T) (Supplementary Fig. 8 and Supplementary Datasets 1–3). While Tes primarily catalyzes the synthesis of GDGT-0[4], GrsA, like Gms, utilizes GDGT-0 as the substrate, but catalyzes the formation of a cyclopentane ring at the C-7 position of GDGT-0[5]. On the other hand, SuiB is a peptide-modifying SPASM-domain-containing RS enzyme that installs a Lys−Trp cross-link in its peptide substrate[17]. These four proteins have different domain compositions but possess similar RS domains (Supplementary Fig. 8). It is

worth noting that Gms and Tes both have an α-domain overlaying above the RS-domain as a 'cap', but they are on opposite sides (Supplementary Fig. 8a, b). This 'cap' and the RS-domain in Tes provide two hydrophobic pockets for substrate binding. This observation suggests that the substrate in Gms may have a different orientation for reaching the RS catalytic center (Supplementary Figs. 7, 8). Although GrsA shares the same substrate as Gms, its structural characteristics diverge notably from Gms (Supplementary Fig. 8d). Firstly, GrsA lacks the 'cap' above the RS domain, which forms an unclosed pocket near its N-domain and RS-domain. This structural model suggests that GrsA does not need the assistance of the 'cap' above the RS-domain to efficiently bind the substrate. Secondly, GrsA has two elongated hydrophobic helices in its C-terminus. Given that the substrate GDGT-0 localizes to the cell membrane, it is conceivable that the β-hairpin of Gms and the C-terminal helix of GrsA may act as anchors to the cell membrane, facilitating potential substrate interactions with these proteins. The structural similarities between SuiB and Gms are pronounced, but their disparate pockets result in distinct substrate affinities and binding specificities (Supplementary Fig. 8a, c). While this comparison between Gms, Tes, SuiB, and GrsA offers insights into substrate binding mechanisms, further experimental validation is needed.

### The biological sources of GMGTs

To identify the biological sources of GMGTs, we performed a BLASTP search against the Non-redundant protein sequences (nr) database in NCBI restricted to archaea genomes for Gms protein homologs. We initially obtained 649 Gms homologs from the BLASTP search (e-value < 1e$^{-50}$, identity >30%, sequence length >420 amino acids), followed by sequence clustering at a 90% sequence identity using CD-HIT tool[19], resulting in 441 homologs. Subsequently, these 441 homologs underwent filtering with sequence similarity networks (SSNs)[20] to identify isofunctional groups of radical SAM enzymes, leading to the identification of 413 Gms homologs based on an alignment score of 90. These 413 homologs are classified within the mega-1-1-24 subgroup and can be distinguished from other identified radical SAM enzymes, such as Tes[3,4] and Grs[5] involved in GDGTs biosynthesis (Supplementary Fig. 9), and AhbC/AhbD[21,22] involved in the anaerobic biosynthesis of heme b in methanogens and sulfate-reducing bacteria (Supplementary Fig. 10). Phylogenetic analysis indicates these 413 Gms homologs are distributed widely in all three archaeal superphyla (Asgard, Tack, and DPANN) and Euryarchaeota (Fig. 3a and Supplementary Fig. 11). Significantly, it is intriguing to note that all Gms homologs are exclusively present in obligate anaerobic archaeal genomes (Fig. 3a).

Lipid analysis of culturable archaea from previous studies was further applied to explore the potential link between GMGTs production (Gms) and anaerobic respiration. Some obligate anaerobic archaea, including *Methanothermobacter thermautotrophicus*, *Palaeococcus helgesonii*, *Pyrococcus furiosus*, *Thermococcus acidaminovorans*, *Aciduliprofundum boonei*, and *Ignisphaera aggregans*, contain Gms homologs and were reported previously to produce GMGTs[8,10,23,24]. In contrast, aerobic archaea, such as marine thaumarchaea, one of the most abundant archaea in marine water columns, do not have a Gms homolog and do not produce GMGTs[25]. Moreover, even the facultative anaerobic archaea, such as *Metallosphaera sedula* and *S. acidocaldarius*, lack the Gms homolog and do not make GMGTs lipids[26,27]. Thus, these observations suggest the biosynthesis of GMGTs is exclusive to obligate anaerobic archaea.

### *gms* gene present in oxygen-deficient environments

The distinct biological source of GMGTs that we have identified suggests that GMGTs producers should be abundant in oxygen-deficient ecosystems and environments. We next investigated the presence of *gms* gene homologs in the metagenomes extracted from oxic and oxygen-deficient environments with tBLASTn searches (e-value < 1e$^{-20}$,

identity >60%, sequence length >80 bp), which indicates the microbial communities have the genetic capacity to synthesize GMGTs. The relative abundances of *gms* were calculated as RPKM (reads per kilobase of sequence per million reads) values. We defined oxygen-deficient environments as $O_2$ < 25 μM, a concentration low enough to induce anaerobic metabolism[28].

Ocean metagenome sequences, in conjunction with oxygen level information, were obtained from the NCBI SRA (Sequence Read Archive) database. These sequences were extracted from the Pacific Ocean, the Black Sea, the Baltic Sea, and the South China Sea blue hole, encompassing various oceanic environments, including the ocean surface, suboxic water (oxygen minimum zone, OMZ), and anoxic marine sediment/water. These metagenome sequences were subsequently analyzed for the presence of *gms*. Consistent with the culture results, *gms* was exclusively detected in suboxic/anoxic (oxygen-deficient) environments, and absent under oxic conditions (Fig. 3b and Supplementary Fig. 12a). However, *gms* was not found in every oxygen-deficient sample. For example, *gms* was not found in certain samples from the Baltic Sea suboxic water column and Black Sea sediments (Fig. 3b). This implies that fluctuations in GMGTs producers within oxygen-deficient environments could be influenced by additional environmental factors such as specific redox conditions.

In addition to the ocean, we observed a comparable distribution pattern of *gms* in freshwater lakes (Fig. 3b and Supplementary Fig. 12b). We specifically chose representative lakes with available metagenome data and oxygen level measurements for our analysis. These lakes included Lake 227 in Canada, Kivu Lake in Rwanda, and Lake Tanganyika in East Africa. Unlike the open ocean, the bottom waters of most lakes are anoxic or even euxinic due to restricted circulation[29]. In these lake environments, *gms* homologs were found exclusively in oxygen-deficient waters. The analysis of metagenomes from both oceans and lakes suggests that GMGTs producers are widespread in a variety of oxygen-deficient environments but absent in oxic environments.

## Discussion

The direct linkage of two sp$^3$ carbons that we propose to occur in the final step of GMGTs biosynthesis is a highly challenging biochemical reaction. Similar bond formation was reported in Tes (or MJ0619)[3,4], and hypothesized for Grs A/B[5]. All of these are radical SAM enzymes, which utilize radical SAM chemistry to accomplish C−C bond formation via two sequential C−H activations on inert sp$^3$-hybridized carbon centers. An important question is how these enzymes are able to retain high-energy carbon radical intermediates, such as the initial species generated via H-atom abstraction by the 5′-dAdo•. For Tes (or MJ0619), an intermediate with a bond between the substrate carbon and sulfur of one auxiliary [4Fe−4S] was identified, providing evidence that the substrate radical is contained by direct coupling with the [4Fe−4S] cluster[4]. However, the radical SAM enzymes GrsA and GrsB that catalyze GDGTs cyclization[5] most likely contain only one radical SAM [4Fe−4S] cluster based on sequence analysis. Therefore, an alternative approach may be used for stabilizing the substrate radical intermediate as there is no auxiliary cluster. In this work, we identified that Gms is a SPASM-domain containing radical SAM enzyme, which harbors one radical SAM [4Fe−4S] cluster and two auxiliary [4Fe−4S] clusters. Therefore, we think it likely catalyzes C−C bond formation by attaching the substrate carbon radical to an auxiliary [4Fe−4S] cluster, similar to Tes.

Why are GMGTs exclusively produced by anaerobic archaea? The Gms enzyme, which belongs to the radical SAM superfamily, catalyzes the reaction under strongly reducing conditions. This fact alone cannot explain why only anaerobic organisms produce GMGTs, as many other radical SAM enzymes are also extensively used by aerobic organisms. The reducing environment within the cell cytoplasm may be sufficient for the radical SAM enzyme to function even under aerobic culture[44]. Alternatively, the formation of the extra covalent

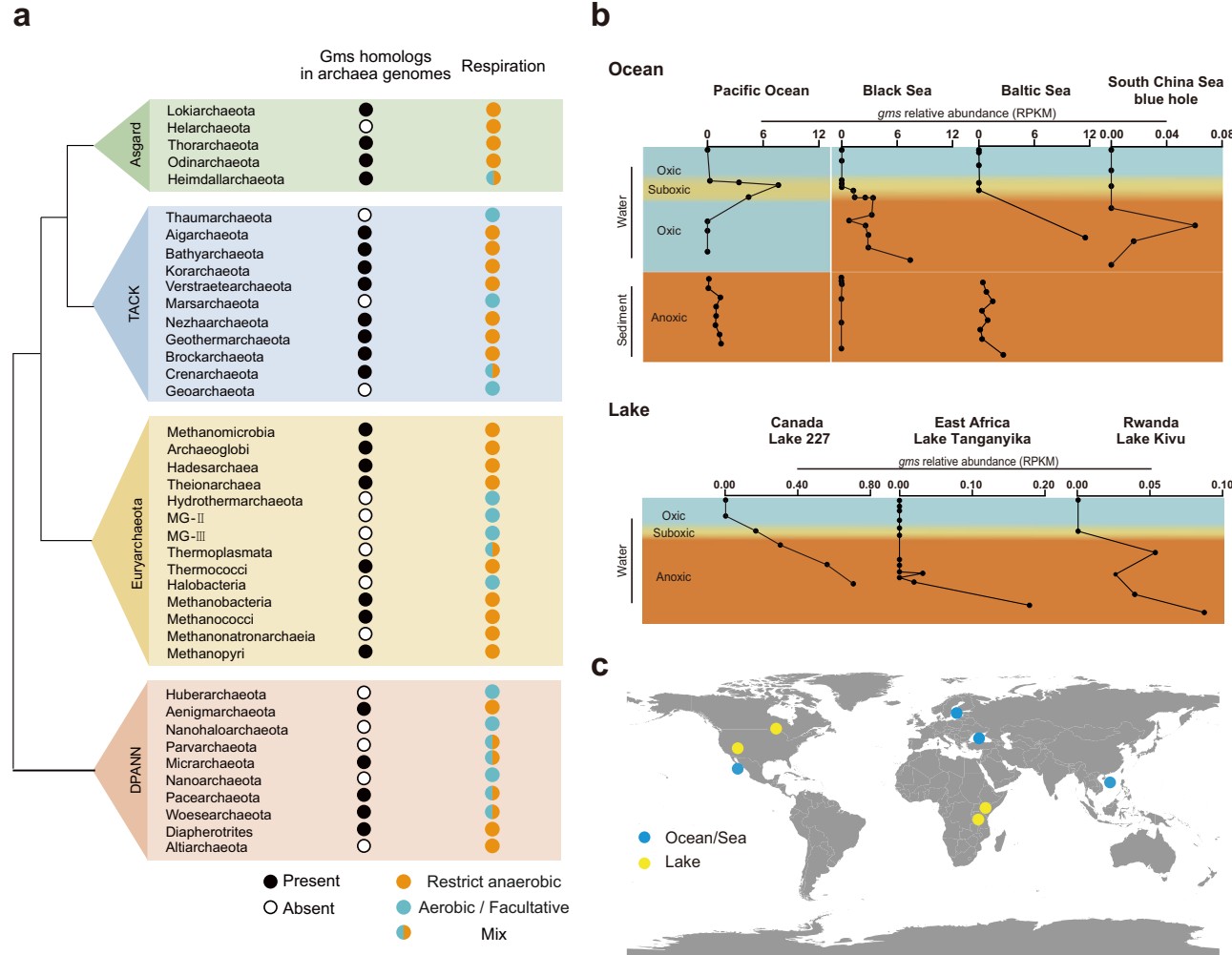

**Fig. 3 | The distribution of Gms homologs in archaea genomes and environmental metagenomes. a** The BLASTp searches of Gms protein homologs in the NCBI database and the respiration features of archaea are derived from published studies[30–42]. Taxonomic groups that contain at least one species with Gms homologs are designated by black circles. It is important to note that the presence of black circles in a group does not imply that every species within the group possesses Gms homologs. Conversely, taxonomic groups lacking any Gms homologs are represented by white circles. For respiration features, the groups containing only obligate anaerobic archaea are marked in orange, aerobic and facultative archaea are marked in blue, and the groups containing anaerobic and aerobic/ facultative archaea are marked in a mix color of orange and blue. **b** The relative abundances of *gms* gene homologs are present as RPKM values, and the reads of *gms* were obtained by tBLASTn searches of metagenomes in the SRA database on NCBI. The sample depths and oxygen concentration profiles are included in the Source Data. The color indicates oxygen levels (blue is oxic, yellow is suboxic, and orange is anoxic). The y-axis represents the approximate depths of water and sediment. **c** Geographical distribution of analyzed metagenome samples. The map was created using Ocean Data View (version 5.6.5)[43]. Source data are provided as a Source Data file.

bond between the two alkyl chains provides extra membrane stability and impermeability[10], and we speculate that this is an adaptive strategy for certain anaerobic archaea to maintain cell integrity under oxidative stress. Further culture studies are needed to interrogate the physiological function of GMGTs under redox stress.

The presence of the *gms* gene exclusively in diverse oxygen-deficient environments, including freshwater and marine ecosystems, suggests GMGTs producers are associated with oxygen-deficient ecological niches. This finding is consistent with results from the phylogenetic analysis, which indicated that GMGTs are synthesized by anaerobic archaea. Results of the geochemical analysis regarding the distribution of GMGTs lipids in geological records will be presented in a separate paper.

In summary, our study has unveiled the biosynthetic gene responsible for GMGTs formation and has identified the unique biological sources of these lipids. Metagenome analysis revealed the exclusive presence of the *gms* gene in oxygen-deficient environments. These findings not only expand our understanding of lipid

biosynthesis and chemistry but also demonstrate the correlation between GMGTs producers and oxygen-deficient habitats.

## Methods

### Microbial strains, media, and growth conditions

Microbial strains used in this study are shown in Supplementary Table 2.

*M. maripaludis* with pMEV4 plasmid was cultured under an anaerobic condition with puromycin (2.5 µg/mL) in McF medium at 37 °C, and the solid medium was prepared by adding 1.5% agar[45]. Lysogeny broth was used to culture *E. coli* DH10B and BL21 (DE3) *ΔiscR* (with kanamycin resistance) at 37 °C.

*S. acidocaldarius ΔgrsAΔgrsB* mutant was grown aerobically at 70 °C, 200 rpm in basal Brock medium supplemented with 0.1% (wt/vol) NZ-amine, 0.2% (wt/vol) sucrose, 10 µg/mL (wt/vol) uracil, and adjusted to pH 3 with sulfuric acid[46]. To cultivate on plates, Brock medium was supplemented with a final concentration of 0.6%–0.8% (w/v) gelrite, 10 mM MgCl₂ and 3 mM CaCl₂.

## Plasmid construction

Plasmids and primers used in this study are shown in Supplementary Tables 3, 4.

Primers were synthesized by Tsingke Company (Beijing, China). PCR was performed using Phusion high-fidelity DNA Polymerase (Vazyme). The Fast Pure Plasmid Mini Kit (Vazyme) was used for the isolation of *E. coli* plasmid DNA. DNA fragments were purified by the FastPure Gel DNA Extraction Mini Kit (Vazyme) for cloning. The ClonExpress One Step Cloning Kit (Vazyme) was used for the construction of the plasmids via stepwise Gibson assembly. DNA sequences were confirmed by sequencing at Tsingke (Beijing, China). In the in vivo studies, to construct the plasmids for the expression in *M. maripaludis*, the *METOK_RSO4425* gene was synthesized by GENEWIZ Company and cloned into the AfeI site of pMEV4-*maeo_0574*.

The plasmid used in the in vitro studies is a C-terminal *strep*-tag II containing pET-21b (+) vector with a codon-optimized *METOK_RSO4425* gene for *E.coli* expression system. To construct the *METOK_RSO4425* mutant plasmids, the *ΔAuxIΔAuxII*, *ΔRSΔAuxII* and *ΔRSΔAuxI* mutants were generated by overlap extension PCR with the corresponding inner primers (Supplementary Table 4) via the stepwise Gibson assembly.

## Transformation of plasmids into host cells

In the in vivo studies, the constructed plasmids were transformed into *M. maripaludis* by polyethylene glycol (PEG)-mediated transformation, screened on puromycin (2.5 µg/mL) in McF agar medium, and verified by PCR amplification with primers P1F/1R[47]. In the in vitro studies, the plasmid was transformed to the electrocompetent cells of *E. coli* DH10B or BL21 (DE3) *ΔiscR* by electroporation using a Micro-Pulser Electroporator (BioRad). The transformants were selected on agar plates with 100 µg/mL ampicillin and 40 µg/mL kanamycin.

## Lipid extraction, purification and analysis

*M. maripaludis* transformants (100 mL) were cultured anaerobically at 37 °C and harvested in the stationary phase. *S. acidocaldarius* *ΔgrsAΔgrsB* mutant was cultured aerobically at 70 °C with shaking at 150 rpm, and harvested in the stationary phase. Cultures were collected by centrifugation at $10,000 \times g$ for 10 min, and pellets were stored at −80 °C before extraction.

The pellets were acid hydrolyzed with 7 mL of 10% (vol/vol) hydrochloric acid (HCl) and methanol (MeOH) in a 40 mL covered glass bottle overnight at 70 °C. 10 mL of dichloromethane (DCM) and 10 mL of pure water were added to the acid-treated mixture, and were transferred to a 50 mL Teflon tube. The samples were centrifuged at $2800 \times g$ for 10 min for separating the aqueous phase from the organic phase. The organic phase was then transferred to the collection glass bottle. The lipids were extracted three times by DCM. Prior to further analysis, the organic phase was combined and filtered through a 0.22 µm PTFE filter, then dried with nitrogen gas[3].

The GDGT-0 was purified from the lipids extracted from the *ΔgrsAΔgrsB* mutant of *S. acidocaldarius* using the preparative LC-MS. GDGT-0 purification was performed on an Agilent 1260 Infinit II HPLC-LC/MSD XT system with a fraction collector. Compound separation was achieved with a C18 column (150 mm × 10 mm, ACE) at 45 °C. Lipids were eluted at 45 °C and for the first 2 min with 100% A and 0% B, where A = MeOH and B = IPA, followed by the following gradient: 90/10 A/B from 2 min to 5 min, 76/24 A/B from 5 min to 15 min, 35/65 A/B from 15 min to 20 min, 10/90 A/B from 20 min to 32 min and finally 100% A from 32 min to the end. The purified GDGT-0 was used as the substrate for in vitro Gms activity assay.

Lipids were analyzed by reversed-phase liquid chromatography-high-resolution mass spectrometry (RP-LC-HRMS). Waters ACQUITY Class I ultra-high performance liquid chromatography (UPLC) is used in combination with a SYNAPT G2-Si quadrupole Time of Flight (qTOF) high-resolution mass spectrometer and an electrospray ionization (ESI) source in positive mode. C18-AR UPLC column (2.1 × 150 mm,

2 µm; ACE) maintained at 55 °C. The total running time was 40 min, and the sample was stored at 7 °C during the running. The flow rate is 0.3 ml/min. The mass spectra were collected by FAST-DDA, MS *m/z* 100–2000 and MS² *m/z* 50–2000. Using the collision-induced dissolution (CID) method, the first 5 ions with the highest intensity and the total ion chromatographic (TIC) threshold > 20000 were fragmented to obtain the product ion spectrum. Real-time dynamic mass exclusion can obtain the MS² spectrum of low intensity ions[48].

## Gms expression and purification

The plasmids of strep-tagged Gms and its variants were transformed into the competent cells of *E. coli* BL21 (DE3) *ΔiscR* with kanamycin resistance. The culture of each recombinant was grown in Luria-Bertani broth with 100 mg/L ampicillin, 40 mg/L kanamycin, 2 mM ammonium ferric citrate, 0.5% (w/v) glucose, and 100 mM 3-(N-morpHolino) propanesulfonic acid (MOPS, pH 8.0) at 28 °C to an $OD_{600} \approx 0.4$. Then the cultures were transferred into the Coy anaerobic chambers, and supplemented with 5 mM Cysteine and 10 mM fumarate. Isopropyl β-D-1-Dthiogalactopyranoside (IPTG) was added to a final concentrate of 0.25 mM to induce protein expression[49]. The cultures were stirred for 22 h, and the cells were harvested by centrifugation at $7800 \times g$ for 30 min at 4 °C. Then the cells were frozen in liquid nitrogen and stored at −80 °C.

The cells were lysed using a Lysis buffer (100 mM HEPES, 150 mM KCl, pH 8.0) containing Bugbuster detergent solution (EMD Millipore), benzonanse (EMD Millipore), rLysozyme (EMD Millipore), and one EDTA free protease inhibitor cocktail tablet (Roche). The cell debris was centrifuged at $29,000 \times g$ at 4 °C for 30 min, and the supernatant was applied to a 50 mL column volume of strep-taction resin. The column was washed with 100 mL buffer W (50 mM HEPES, 150 mM KCl, pH 8.0) to remove nonspecific binding proteins. The protein was then eluted using 80 mL buffer W containing 3 mM desthiobiotin, and the dark brown protein fraction was collected. The as-eluted protein, with a concentration of ~300 µM, was aliquoted, flash frozen and stored in liquid nitrogen.

## EPR spectroscopy and analysis

Gms reaction mixtures were prepared anaerobically. 100 µL of as-eluted Gms (~300 µM) or Gms variants (~300 µM) in buffer W solution was incubated with 40 equiv. of sodium dithionite (DTH) or/and 40 equiv. of SAM. Samples were frozen and stored in liquid nitrogen for EPR characterization.

X-band (9.35 GHz) CW EPR spectra were recorded on a Bruker EleXsys E500 spectrometer equipped with a super-high Q resonator (ER4122SHQE) in the Department of Chemistry at the Southern University of Science and Technology (SUSTech). Cryogenic temperatures were obtained and controlled by using an ESR900 liquid helium cryostat in conjunction with a temperature controller (Oxford Instruments Mercury iTC) and a gas flow controller. CW EPR spectra were recorded at 10 K using the following parameters: microwave power = 0.02 mW; conversion time = 40 ms; modulation amplitude = 0.05 mT; modulation frequency = 100 kHz. Simulations of the CW spectra and the following pulse EPR spectra were performed using EasySpin 5.2.35 toolbox[50,51] within the Matlab 2014a software suite (The Mathworks Inc., Natick, MA).

## Structure prediction

The Gms, Tes, and GrsA structures were predicted in AlphaFold2[16]. Because the structure of GrsA does not exist in the AlphaFold2 protein structure database (https://alphafold.ebi.ac.uk), to ensure parameter consistency, we predicted all three proteins using a standalone platform, which was set up on a local computer running the CentOS operating system, and accelerated by one NVIDIA GeForce RTX 2080 Ti GPU with the parameters set as MAX_TEMPLATE_HITS = 20, RELAX_MAX_ITERATIONS = 0, RELAX_ENERGY_TOLERANCE = 2.39,

RELAX_STIFFNESS = 10.0, RELAX_MAX_OUTER_ITERATIONS = 20. Five models were predicted and ranked based on the pLDDT values, representing different levels of confidence. Finally, five models with similar interaction modes were generated, and rank_0 was used for further analysis. Compared to AF-F8AME3-F1-model_v4 and AF-Q4JAU6-F1-model_v4 in the AlphaFold database, the RMSD (root-mean-square deviation) values of Gms and Tes are 0.178 and 0.220, respectively, indicating a proper structure prediction. The predicted model of Tes showed a similar structure to the published homolog, GDGT-MAS (PDB: 7TOM, RMSD = 0.992). Since GDGT-MAS had a C-terminal missing in the structure, we here used the predicted model of Tes for comparison.

## Gms activity assay

The purified GDGT-0 was resuspended in 20 μL DMSO and 100 μL of buffer W, by incubating the solution at room temperature for hours. The reaction solution contained 25 μM Gms, 30 equiv. of DTH, 30 equiv. of SAM and 10 equiv. of GDGT-0 in 1.5 mL buffer W at 37 °C for 72 hr. An aliquot of the reaction solution was extracted by 10% (vol/vol) hydrochloric acid (HCl) in methanol (MeOH) for lipid analysis. The activity assays were carried out in triplicate.

## Bioinformatic analyzes

BLASTP searches of Gms homologs were conducted in the Non-redundant protein sequences (nr) database in NCBI with METOK_RS04425 amino acid sequence from *M. okinawensis* as the search query. Default settings were used for all searches. The searches were restricted to Asgard (taxid:1935183), TACK (taxid:1783275), Euryarchaeota (taxid:28890), DPANN (taxid:1783276), respectively, to identify the Gms homologs (e-value < 1e$^{-50}$, identity >30%, length >420 aa). Redundant sequences were removed by clustering at a 90% sequence identity using CD-HIT version 4.6.8[19]. SSNs analysis was permformed by RadicalSAM.org[20] with an alignment score of 90 to identify radical SAM protein isofunctional groups. Twenty archaeal GrsA sequences were used as the outgroup. Protein sequences were aligned via MUSCLE version 3.8.31[52] with the outgroup included. Maximum likelihood trees were built by RAxML version 8.2.12[53] using the CAT model of rate heterogeneity with the LG substitution matrix and 1000 bootstrap iterations. interactive Tree Of Life (iTOL) was used for tree visualization and annotation (http://itol.embl.de/)[54].

To search *gms* gene homologs in metagenomes, the program NCBI-BLAST 2.2.27+ was used for TBLASTN. Default settings were used for all searches. The *gms* homologs were identified by using TBLASTN searches (e-value < 1e$^{-20}$, identity >60%, length >80 bp) of the SRA (Sequences Read Archive) database on NCBI with the METOK_RS04425 DNA sequence from of *M. okinawensis* as the search query. The GenBank metagenomes accessions for searches in Fig. 3 are listed in Source Data.

## Reporting summary

Further information on research design is available in the Nature Portfolio Reporting Summary linked to this article.

# Data availability

All data are available in the manuscript or the Supplementary Information file, Supplementary Datasets, and Source Data files. Source data are provided in this paper.

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

## Acknowledgements

We thank Weiqi Yao and Lu Fan from the Southern University of Science and Technology, Xiao-lei Liu from the University of Oklahoma, Jérôme Kaiser from Leibniz Institute for Baltic Sea Research – Warnemünde (IOW), B. David A. Naafs from the University of Bristol, Liang Dong and Fengping Wang from Shanghai Jiao Tong University, and Kai-Uwe Hinrichs from the University of Bremen for constructive discussion. We also thank Miao Huang from the China University of Geosciences (Wuhan) and Fengfeng Zheng from the Southern University of Science and Technology for the help in lipid analysis. F.Z. is an investigator of SUS-Tech Institute for Biological Electron Microscopy. Support for this study was provided by the National Nature Science Foundation of China to Z.Z. (Nos. 92351301, 32393974 and 32170041), Shenzhen Science and Technology Program to L.T. (No. 20231120094247002), Guangdong Innovative and Entrepreneurial Research Team Program to F.Z. (No. 2021ZT09Y104), and National Science Foundation to N.Z.B. (CAREER CHE-1846512).

## Author contributions

Z.Z., L.T. and F.Z. were responsible for the conceptualization and supervision of the project; Y.L., L.T., T.Y., F.Z., B.Z., H.C., H.R.A. and N.Z.B. conducted genetic, biochemical and structural experiments; X.F., Y.L., X.C., X.Z. and Z.Z. performed bioinformatic analysis; Y.L. and H.Y. performed lipid extraction and analysis; Y.L., Z.Z., L.T. and F.Z. wrote the manuscript with input from all of the authors.

## Competing interests

The authors declare no competing interests.
