## [Peer Review File · Nature Communications]

Biosynthesis of GMGT lipids by a radical SAM enzyme associated with anaerobic archaea and oxygen-deficient environmentsReviewer #1 (Remarks to the Author):

Review Li et al., Nature Comm.

This manuscript deals with various aspects (biosynthesis, enzymology, physiology, biogeochemistry and molecular paleontology) of so-called isoprenoid GMGTs, archaeal membrane lipids that are produced by "cross-linking" of the much more common GDGTs. These GMGTs have been reported before in archaea but the exact chemical structure and biosynthetic pathway by which GDGTs are produced and potential biomarker value remains elusive. The novel aspect of this manuscript is that it for the first time (although a similar manuscript from the group where the last and corresponding author performed a post-doc from 2015-2019 has been posted on a preprint server; <https://www.biorxiv.org/content/10.1101/2023.10.20.563219v1.abstract>) identifies the gene encoding the enzyme responsible for the "crosslinking". This allowed to identify the presence of this gene in archaeal (meta)genomes, revealing that its occurrence is limited to anaerobic archaea, which is suggested to result in a biomarker potential of isoprenoid GMGTs.

Although I feel the microbiological work is very interesting and well executed, there are several major issues that prevent me to recommend this manuscript for publication in Nature Communications.

1) It starts with the nomenclature: the authors call the archaeal membrane lipid of interest H-GDGT-0, which is chemically incorrect. When the two biphytanyl chains are cross-linked the GDGT becomes a glycerol MONOalkyl glycerol tetraether (GMGT); this is NOT a GDGT and this should be reflected in its name and, consequently, in the name of the enzyme and gene encoding the enzyme (the H in Hbs does not make sense). If you look to the "general" structure of GMGT-0, the introduction of the bridge actually results in a structure that looks like an eight (8; two macrocycles connected) not an H-structure (that only holds when you break the four ether linkages). I guess the origin of the use of H in the name is the publication of Morii et al. *Biochim Biophys Acta* 1390, 339-345 (1998) but there the authors specifically refer to the H-shaped core of the membrane lipid.

2) I find it peculiar that the authors have done a lot of experiments to establish the enzyme producing a biochemical product that has not been fully characterized chemically. Lutnaes et al. (2007) elucidated the structure of a series of octaterpene tetracarboxylic acids with 4-6 cyclopentane moieties in petroleum by two-dimensional NMR studies. This established the position of the bridge, i.e. between two mid-chain methyl groups of each biphytanyl chain (although this still leaves various isomers possible). These tetracarboxylic acids were interpreted to be diagenetic products of isoGMGTs derived from hyperthermophilic archaea. Hence, direct structural identification of isoGMGTs in cultures of archaea is still lacking; the assignment of the position of the C-C bridge in their presumed diagenetic products by Lutnaes et al. (2006, 2007) is currently the best available indirect evidence. The authors do not provide any other chemical evidence other than showing a mass spectrum revealing a molecular ion that is two daltons lower than that of GDGT-0 (actually the spectra they show in Fig. 1C have been published previously and there is no good reason to show them again) and fail to acknowledge this weakness. In fact, a substantial part of the manuscript (lines 121-169) deals with details of GMGT-0 biosynthesis with predictions of the active sites of the enzyme (see Extended Data Fig. 6, where specifically two methyl groups (in red) of GDGT-0 are indicated to enter the active site) without knowing with certainty what the structure of the product of the enzyme reaction is. I find this unacceptable: the authors should isolate GMGT-0 from one of their cultures and prove the chemical structure they propose by 2D-NMR or related techniques. In the abstract they claim "This marks the second known example of direct linkage of two sp³ carbons by archaea", but in fact we are not sure of the structure of the product produced and thus the sites of crosslinking, so this is highly speculative. The authors have isolated the enzyme and performed an in vitro study, which is very nice, but they should have isolated the product produced by this enzyme as well.

3) The finding that the enzyme responsible for the production of GMGT-0 is restricted to anaerobic archaea is interesting but it remains unclear how they have established this since many of the investigated species seem not to be available in culture (species names in Extended Data Fig. 7 are unreadable; many seem to belong to groups that contain no cultured relative). Also, it remains entirely unclear what the role of temperature is in the formation of GMGT-0. This membrane lipid has mostly been reported to occur in (hyper)thermophilic archaea and the "bridging" has been

proposed as an adaptation to temperature to rigidify the membrane. This is not discussed at all. Furthermore, the authors do not provide a plausible explanation why "bridging" of GDGTs would be an advantage for an archaeon in an anaerobic environment. They hypothesize that it could be an adaptation to oxidative stress but fail to present data to support that. Furthermore, this would be in contradiction with their proposal that GMGT-0 is a "novel biomarker for anoxia".

4) The quest for the "hbs gene" in the environment generates questions. It remains unclear why different tBLASTn criteria (lines 194) were used as compared to those (lines 172-173) in the search in archaeal genomes. Especially the use of a sequence length of only >80 bp (equivalent to >27 aa?) seems risky. But I am not an expert in this field, so I might be wrong.

5) The environmental dataset on GMGT-0 presented are not convincing. Firstly, the relative abundance of GMGT-0 (it is actually not clear if this is the only GMGT taken into account or if this includes GMGTs with cyclopentane moieties) is extremely low. In the SPM of the Baltic Sea and Black Sea it is never >0.2% of the total GDGTs (I guess they mean total isoprenoid GDGTs). This is extremely low and, in fact, generates analytical challenges to assess their relative amount. The profile of the Black Sea also shows the presence in oxic waters and a decrease in the euxinic waters. Actually, the Black Sea is 2200 m deep and the sampling stops at 110 m, so this sample set cannot reveal what is going on in this euxinic basin since only 5% of the depth range is covered. The SPM profile of the Baltic Sea contains only four datapoints and the two in the sub-oxic zone shows the highest relative abundance, whilst there is only one sample in the oxic zone obtained from the surface where the abundance of archaeal GDGTs is typically below detection limit. Hence, both SPM profiles do not contribute to the story. The data from modern wetlands shows indeed a higher relative abundance (ca. 4%) of GMGT-0 in anoxic soil but this dataset has been published in 2018 by Naafs et al., so this is telling nothing new ("we suggest that H-GDGTs are preferentially produced at depth, in the anoxic, water-saturated catotelm by anaerobic archaea and bacteria. This is consistent with the observation that all reported organisms that produce H-isoGDGTs are strict anaerobes"; quote from Naafs et al.). The data presented also only provide the relative abundance of GMGT-0 whereas absolute concentrations (normalized on TOC) would be required; changes in the absolute abundance of GDGTs now also affect the relative abundance of GMGT-0. In conclusion, the environmental datasets do not contribute anything convincing or new and should be left out.

6) The whole story is framed as the data would provide a novel biomarker "for studying ocean anoxic events dating back hundreds of millions of years in Earth's history" (see also the start of the introduction). I don't see the advantage of using GMGT-0 in the study of OAEs. We already have many biomarkers to indicate that the water column has been anoxic (e.g. dinorhopane, lycopene etc.) and some of these even allow us to assess that the lower part of photic zone was euxinic (e.g., isorenieratene derivatives; Sinninghe Damste and Koster, 1998; Kuypers et al., 2002). What is GMGT-0 offering in addition, assuming we are convinced it can only originate from anaerobic archaea? The authors already acknowledge themselves that they can't discriminate between archaea residing in the water column or in the sediment (that ultimately will become anoxic upon burial). So, what is going to reveal this in addition to what all the biomarker data on OAEs that have been obtained in the last 4-5 decades? I don't see the point and it is also not provided in the manuscript. If we look to the data of OAE2 (Demerara Rise), we again see that GMGT-0 (perhaps including GMGTs with cyclopentane moieties; no information provided) is again present in very low relative abundance (ca. 0.5% on average). This strongly contrasts the statement in the abstract "remarkably abundant in ancient sediments during past periods of oxygen deficiency, such as OAE-2". A slight increase is seen during OAE-2 but at this site bottom waters were probably also anoxic before and after the OAE, otherwise, it would have been difficult to explain TOC levels of 10% throughout this section. This is also concluded on the basis of more established biomarkers in a recent paper (Abraham et al., 2023), which is co-authored by one of the co-authors of this manuscript. So, it remains totally unclear what the GMGT-0 profile is indicating.

7) The scientific contribution of some of the authors also raises questions. The author contribution mentions "X.L., H.Y., J.K., G.T.C., A.K.W., B.D.A.N., L.D. and F.W. provided environmental samples and performed lipid analysis" but as far as I can see there is only data of three co-authors: JK and AKW (environmental GDGT and GMGT data Black Sea and Baltic Sea; Fig. 4a) and BDAN (GDGT and GMGT data Modern Wetland (Fig. 4b) and OAE2 Demerara Rise (Fig. 4c). The GDGT data for these sites have previously been published in Wittenborn et al (2023), Naafs et al. (2018), and Abraham et al. (2023). The relative quantitation of GMGT-0 in these cases only required the integration of one additional peak in their dataset (in case of Naafs et al., 2018 this was already reported). So, it remains unclear why there is such an extensive experimental description of

GDGT/GMGT analysis in this manuscript; the authors could have just referred to these paper. The contribution of the other five authors remains a mystery. There is a reference to a paper of Connock et al. (ref 45) in the legend of Fig. 4 but this paper refers to a site in the South Atlantic and not in the proto North Atlantic Ocean where Demerara Rise is located.

I would strongly suggest focusing on the strong point of the paper (the biochemical characterization of the enzyme and apparent limitation to anaerobic archaea) and leave the rest out.

More detailed comments:

Line 1 Title doesn't read well. Furthermore, the authors do not provide any mechanism why GMGT-0 is only produced in anaerobic archaea. The title should reflect what has really been demonstrated (e.g. "Anaerobic archaea produce GMGT-0 by a novel biphytanyl crosslinking enzyme").

Line 30 "Often overlooked subset" Why?

Line 31 Clumsy writing: it is the enzyme that is primarily responsible. This sentence can be written in a more clear way.

Lines 41-44 Very speculative and not supported by the data provided.

Lines 47-70 The introduction has no focus and is poorly written (e.g. no connection between paragraph #1 and #2).

Line 55 If you want to refer to a paper that applies GDGT for SST reconstruction during OAEs an original paper needs to be quoted. Ref 4 does not make sense in this respect.

Line 66 "extra cross link"? How do you mean? GDGT have no cross-link

Line 69 "synthetic mechanisms". Unclear what this means.

I have only annotated page 1-2. This paper has 20 authors, some are based on highly ranked universities where English is the mother tongue. I can hardly believe that all of them have thoroughly gone through the text and made scientific and editorial comments. It is not the task of a referee to do this job.

Jaap Sinninghe Damsté

February 5, 2024

Abraham, M. A. F., Naafs, B. D. A., Lauretano, V., Sgouridis, F., & Pancost, R. D. (2023). Warming drove the Expansion of Marine Anoxia in the Equatorial Atlantic during the Cenomanian Leading up to Oceanic Anoxic Event 2. *Clim. Past*, 19, 2569–2580.

Kuypers, M. M., Pancost, R. D., Nijenhuis, I. A., & Sinninghe Damsté, J. S. (2002). Enhanced productivity led to increased organic carbon burial in the euxinic North Atlantic basin during the late Cenomanian oceanic anoxic event. *Paleoceanography*, 17(4), 3-1.

Lutnaes, B. F., Krane, J., Smith, B. E., & Rowland, S. J. (2007). Structure elucidation of C 80, C 81 and C 82 isoprenoid tetraacids responsible for naphthenate deposition in crude oil production. *Organic & Biomolecular Chemistry*, 5(12), 1873-1877.

Naafs, B. D. A., McCormick, D., Inglis, G. N., & Pancost, R. D. (2018). Archaeal and bacterial H-GDGTs are abundant in peat and their relative abundance is positively correlated with temperature. *Geochimica et Cosmochimica Acta*, 227, 156-170.

Sinninghe Damsté, J. S., & Köster, J. (1998). A euxinic southern North Atlantic Ocean during the Cenomanian/Turonian oceanic anoxic event. *Earth and Planetary Science Letters*, 158(3-4), 165-173.

Wittenborn, A. K., Bauersachs, T., Hassenrück, C., Käding, K., Wäge-Recchioni, J., Jürgens, K., ... & Kaiser, J. (2023). Nitrosopumilus as main source of isoprenoid glycerol dialkyl glycerol tetraether lipids in the central Baltic Sea. *Frontiers in Microbiology*, 14.

Reviewer #2 (Remarks to the Author):

This manuscript is about the identification of the enzyme responsible of converting the membrane-spanning tetra ether lipid GDGT-0 into the bridged derivative H-GDGT-0. Through heterologous expression and in vivo lipid analysis, as well as protein purification and in vitro enzyme assays, the authors convincingly show that this C-C bonding enzyme belongs to the radical SAM class of enzymes, elucidating a further enzymatic step in archaeal lipid biosynthesis.

comments

Fig. 1b, the in vitro activity of the purified enzyme seems very low. Likely the enzyme readily inactivates when exposed to oxygen; or cofactor incorporation might be very poor. The authors should consider the optimised conditions for radical SAM protein expression as described in ref 6. This is important as now it is not entirely clear if the further analysis (EPR) was done on a fully functional enzyme.

Otherwise, in this in vitro assay, it could be an issue of substrate exposure to the enzyme. I noticed the assays do not contain detergent which would be necessary to keep the GDGT-0 monodispersed.

The authors propose a putative mechanism using the alfaFold2 model and insights on other radical SAM proteins. However, what I entirely miss in the discussion is how this enzyme can act on a substrate that supposedly is membrane spanning. How does the binding site even get access to the relevant part of the substrate, that in the membrane spanning state will be in the middle of the membrane! A transparent discussion is needed.

Reviewer #3 (Remarks to the Author):

Glycerol dialkyl glycerol tetraethers (GDGTs) are a special class of archaeal ether lipids that are membrane-spanning and can have several modifications/variations like crosslinks between the isoprenoid chains (H-GDGTs) or rings within the isoprenoid chains. Only in the last few years, the discovery of the enzymes that catalyze the reactions at the isoprenoid chains began (Tes, GrsA, GrsB), they are all radical SAM (RS) enzymes. The presented manuscript is a continuation of the elucidation of these interesting enzymes, as it describes for the first time the RS enzyme that introduces the crosslinks in H-GDGTs, why the authors call it "H-GDGT bridge synthase" (Hbs).

The authors study the Hbs activity in vitro and in vivo. Furthermore, they explore the occurrence of Hbs homologues in fully sequenced species and in metagenomic data. They also provide new own data on the occurrence of H-GDGTs in oxygen-deficient and oxic water samples. From all those data, they deduce that H-GDGT producing species are obligate anaerobic, and therefore only occur in (almost) oxygen-free environments. They suggest that H-GDGTs might be suitable as a new biomarker for studying oxygen deficiency in the geological past.

The study is, in most parts, properly done, and the results are properly presented. However, I have one main concern. Minor issues are listed below.

Main concern:

The authors claim that Hbs homologues only occur in obligate anaerobic species (l. 171 ff), which is stressed as one of the major conclusions from this study. For this, they do BLAST searches within archaeal genomes with defined search parameters (e-value limit, min. identity, ...). The authors do, however, not prove that these parameters are suited to distinguish between "true" Hbs and other related RS enzymes like for example Tes that also occur in aerobic species. The authors must somehow show that the hits they get are homogenous in their sequence properties, but different from Tes, Grs, or other, not yet characterized RS enzymes. This could, for example, be done by using the sequence similarity network (SSN) technique, or at least by showing adequate

multiple sequence alignments or phylogenetic trees with all these RS enzymes (not only Grs as an outgroup as in ext. data Fig. 7). The same applies for the search in the metagenomes. For clarity, the authors should not only show the hits among metagenomic data in source data (is this really what is shown according to l. 606 in "Archaeal domain-Hbs-homologs"?) but also all the hits from the BLASTP search.

In general (see l.130ff, and also 1st paragraph of discussion), the whole story would benefit from a more detailed description of the structural properties of Hbs in comparison with Tes (which also has auxiliary Fe-S clusters), GrsA/B (which have no auxiliary Fe-S clusters) and SuiB. This could be done as extensions to Fig. 1 and ext. data Fig. 1, the EPR data then could be a separate figure.

Other issues, in order of appearance:

Carefully read again the manuscript for good language and typos, especially ll. 530-545 are full of them.

ll. 78-81: The result of this comparison should be better presented

l. 82f: a gene cannot contain a motif defined by amino acids, only a protein can contain such a motif

l. 121 ff: As EPR is a rarely seen method, it might be helpful for readers not so familiar with it to very briefly resume at the beginning of the paragraph what can be measured with this method.

ext. data Fig. 1: At the time of the first reference to this figure in the text, nobody knows yet that WP_62369926.1 is Hbs, so both names should be used in the figure legend. It must also be introduced in the legend that the structure is a model, as also this is only introduced later in the text.

Fig. 2: The language of the legend should be improved (e.g.: "a, Four domains shown in Hbs").

l. 172/173: a) BLASTP can only search in protein databases, but not in "archaeal genomes" = DNA sequences; b) Is it really the case that the search was limited to archaea? This does not become clear from the according methods section. What about hits in bacteria? This might be interesting for some people because Tes homologues also occur in bacteria.

l. 174: the authors state that they got 649 hits. In methods, they state that they cluster sequences with 90% similarity, in ext. data Fig. 7 they mention 177 unique homologues from 910 genomes. How do those numbers fit together, especially the 910 genomes?

ext. data Fig. 7: "Based on 177 unique archaea Hbs homologues and 19 archaea Grs homologues from 910 archaea genomes in JGI IMG 691 database as outgroups [...]": the sentence as a whole seems strange: are the 177 unique homologues from the 910 genomes, or the 19 Grs sequences, or both together? Why more genomes than sequences?

ext. data Fig. 7: I cannot see any blue circles as mentioned in the legend.

ext. data Fig. 7: It would be appropriate to list also the Grs sequences in "source data". With respect to my main concern: Would Tes as an outgroup also work?

l. 212: ext. data Fig. 8b? 8c does not exist.

Fig. 3b, c: What is the y-axis of these diagrams? Water/Sediment depth like in Fig. 4?

l. 249ff: Shouldn't equation 1 better already appear two paragraphs above (ll. 236-241)? Or move it to methods?

l. 308f: "The presence of H-GDGTs primarily depends on shift in its associated microbial communities in the water column or sediment, which are highly sensitive and responsive to the

rise of anoxic condition.”: I do not completely understand the meaning of this sentence. Better: ... depends on the distribution of the species that produce the H-GDGTs?

l. 312 f.: “[...] the H-GDGTs proxy has the advantage to yield a more precise and regional representation of ocean oxygen deficiency supported by its well-established biological basis.” I also do not understand this completely. What is meant with “its well-established biological basis”?

l. 539 and elsewhere: “centrifugation at 6,000 rpm for 30 min at 4 °C”: it makes only sense to give the rpm if you also mention the rotor. Alternatively, give the g value.

l. 593: is really clustered at 90% sequence similarity (using a scoring matrix) or rather at 90% sequence identity?

Ext. data Table 1: separating horizontal lines between the *M. maripaludis*, *E. coli* and *S. acidocaldarius* strains might improve clarity. Typo “*Escherichia coil*”.

Point-by-Point Response

Reviewer #1 (Remarks to the Author):

Review Li et al., Nature Comm.

This manuscript deals with various aspects (biosynthesis, enzymology, physiology, biogeochemistry and molecular paleontology) of so-called isoprenoid GMGTs, archaeal membrane lipids that are produced by “cross-linking” of the much more common GDGTs. These GMGTs have been reported before in archaea but the exact chemical structure and biosynthetic pathway by which GDGTs are produced and potential biomarker value remains elusive. The novel aspect of this manuscript is that it for the first time (although a similar manuscript from the group where the last and corresponding author performed a post-doc from 2015-2019 has been posted on a preprint server; <https://www.biorxiv.org/content/10.1101/2023.10.20.563219v1.abstract>) identifies the gene encoding the enzyme responsible for the “crosslinking”. This allowed to identify the presence of this gene in archaeal (meta)genomes, revealing that its occurrence is limited to anaerobic archaea, which is suggested to result in a biomarker potential of isoprenoid GMGTs.

Although I feel the microbiological work is very interesting and well executed, there are several major issues that prevent me to recommend this manuscript for publication in Nature Communications.

We thank Dr. Damsté for the insight review and valuable comments. In response to his suggestions, we have decided to remove the environmental GMGT lipid data and instead focus on the rest aspects of our study.

1) It starts with the nomenclature: the authors call the archaeal membrane lipid of interest H-GDGT-0, which is chemically incorrect. When the two biphytanyl chains are cross-linked the GDGT becomes a glycerol MONOalkyl glycerol tetraether (GMGT); this is NOT a GDGT and this should be reflected in its name and, consequently, in the name of the enzyme and gene encoding the enzyme (the H in Hbs does not make sense). If you look to the “general” structure of GMGT-0, the introduction of the bridge actually results in a structure that looks like an eight (8; two macrocycles connected) not an H-structure (that only holds when you break the four ether linkages). I guess the origin of the use of H in the name is the publication of Morii et al. *Biochim Biophys Acta* 1390, 339-345 (1998) but there the authors specifically refer to the H-shaped core of the membrane lipid.

We included both “H-GDGT” and “GMGT” terminologies in our manuscript. The reason we used “H-GDGT” as the primary term is its accessibility for general audiences and its association with GDGT molecules as a derivative. This study showing GDGT is the substrate for H-GDGT through cross linking of isoprenoid chains in GDGT. Initially, we considered “H-GDGT,” while less chemically precise than “GMGT,” to hold value for general audiences due to its straightforward recognition and its direct link to GDGT molecules.

We understood the reviewer's concerns and have made the necessary adjustments. Thus,

we have used “GMGT” as the primary term, and renamed the enzyme as “GMGT synthase (Gms)” in manuscript.

2) I find it peculiar that the authors have done a lot of experiments to establish the enzyme producing a biochemical product that has not been fully characterized chemically. Lutnaes et al. (2007) elucidated the structure of a series of octaterpene tetracarboxylic acids with 4–6 cyclopentane moieties in petroleum by two-dimensional NMR studies. This established the position of the bridge, i.e. between two mid-chain methyl groups of each biphytanyl chain (although this still leaves various isomers possible). These tetracarboxylic acids were interpreted to be diagenetic products of isoGMGTs derived from hyperthermophilic archaea. Hence, direct structural identification of isoGMGTs in cultures of archaea is still lacking; the assignment of the position of the C–C bridge in their presumed diagenetic products by Lutnaes et al. (2006, 2007) is currently the best available indirect evidence. The authors do not provide any other chemical evidence other than showing a mass spectrum revealing a molecular ion that is two daltons lower than that of GDGT-0 (actually the spectra they show in Fig. 1C have been published previously and there is no good reason to show them again) and fail to acknowledge this weakness. In fact, a substantial part of the manuscript (lines 121–169) deals with details of GMGT-0 biosynthesis with predictions of the active sites of the enzyme (see Extended Data Fig. 6, where specifically two methyl groups (in red) of GDGT-0 are indicated to enter the active site) without knowing with certainty what the structure of the product of the enzyme reaction is. I find this unacceptable: the authors should isolate GMGT-0 from one of their cultures and prove the chemical structure they propose by 2D-NMR or related techniques. In the abstract they claim “This marks the second known example of direct linkage of two sp³ carbons by archaea”, but in fact we are not sure of the structure of the product produced and thus the sites of crosslinking, so this is highly speculative. The authors have isolated the enzyme and performed an in vitro study, which is very nice, but they should have isolated the product produced by this enzyme as well.

Purification of milligrams of lipid substrates GDGT-0 and the product GMGT-0 is a challenging and time-consuming task, which are not commercially available. Despite our numerous efforts to purify these compounds, we have encountered difficulties in obtaining a sufficient amount for NMR analysis. Additionally, the NMR data presented by Lutnaes et al. (2006, 2007) already provided clear evidence for the position of cross-linkage. Thus, we believed that the NMR characterization of GMGT required additional study that is beyond the scope of this manuscript.

The Fig. 1c has been removed to supplementary as requested. The spectra in Fig. 1c were used to show the identification of GMGT compound in our samples, rather than to declaim the novelty of these spectra.

References

1. Lutnaes, Bjart Frode, et al. Archaeal C 80 isoprenoid tetraacids responsible for naphthenate deposition in crude oil processing. *Organic & biomolecular chemistry* 4.4 (2006): 616-620.
2. Lutnaes, Bjart F., et al. Structure elucidation of C 80, C 81 and C 82 isoprenoid tetraacids responsible for naphthenate deposition in crude oil production. *Organic & Biomolecular*

Chemistry 5.12 (2007): 1873-1877.

3) The finding that the enzyme responsible for the production of GMGT-0 is restricted to anaerobic archaea is interesting but it remains unclear how they have established this since many of the investigated species seem not to be available in culture (species names in Extended Data Fig. 7 are unreadable; many seem to belong to groups that contain no cultured relative). Also, it remains entirely unclear what the role of temperature is in the formation of GMGT-0. This membrane lipid has mostly been reported to occur in (hyper)thermophilic archaea and the “bridging” has been proposed as an adaptation to temperature to rigidify the membrane. This is not discussed at all. Furthermore, the authors do not provide a plausible explanation why “bridging” of GDGTs would be an advantage for an archaeon in an anaerobic environment. They hypothesize that it could be an adaptation to oxidative stress but fail to present data to support that. Furthermore, this would be in contradiction with their proposal that GMGT-0 is a “novel biomarker for anoxia”.

A new phylogenetic tree figure (Supplementary Fig. 9) has been added, with improved figure resolution.

GMGTs analysis from peatlands suggested a positive correlation between the relative abundance of GMGTs and temperature (Naafs et al., 2018). In contrast, culture experiments with the thermophilic archaeon *Pyrococcus furiosus* indicated a negative correlation between GMGT abundance and temperature (Tourte et al., 2022), suggesting that GMGT formation may not primarily be a response to heat stress, but rather to other environmental factors. Furthermore, our homolog search for Gms revealed the genetic capability of many mesophilic anaerobic archaea, such as Bathyarchaea, to produce GMGTs, despite not experiencing heat stress. Hence, in this study, we do not consider temperature as the primary factor influencing GMGT formation; instead, we propose that anoxic conditions and oxidative stress play more significant roles. These statements have been incorporated into the Introduction section (line 50 to 64).

We found that GMGTs were produced by anaerobic archaea, which mainly live in anoxic environments. We also observed that the abundance GMGTs lipids were more enriched in suboxic zone than anoxic zone in modern sea (previous Fig. 4A). Based on these observations, we hypothesized that trace of oxygen in suboxic environments, such as oxic-anoxic fluctuation environments, maybe stimulate GMGTs producers to produce more GMGTs to tolerate oxidative stress. Therefore, this hypothesis aligns with our proposal that GMGTs serve as a potential proxy for oxygen deficiency, without contradiction.

References

1. Naafs, B. D. A., et al. "Archaeal and bacterial H-GDGTs are abundant in peat and their relative abundance is positively correlated with temperature." *Geochimica et Cosmochimica Acta* 227 (2018): 156-170.
2. Tourte, Maxime, et al. "Membrane adaptation in the hyperthermophilic archaeon *Pyrococcus furiosus* relies upon a novel strategy involving glycerol monoalkyl glycerol tetraether lipids." *Environmental Microbiology* 24.4 (2022): 2029-2046.

4) The quest for the “hbs gene” in the environment generates questions. It remains unclear

why different tBLASTn criteria (lines 194) were used as compared to those (lines 172-173) in the search in archaeal genomes. Especially the use of a sequence length of only >80 bp (equivalent to >27 aa?) seems risky. But I am not an expert in this field, so I might be wrong.

Two different methods were used for the search of Gms protein/gene homologs in archaeal genomes and environmental metagenomes, respectively.

For the search of protein homologs in archaeal genomes, the full length of the Gms protein amino acid sequence was used as a query to conduct a BLASTP search in NCBI archaeal genomes. (e-value < $1e^{-50}$, identity > 30%, sequence length > 420 amino acids).

However, to quantify the abundance of *gms* gene homologs in environmental metagenomes, a different search method was used. This is because the raw data of metagenomes in the SRA (Sequence Read Archive) database consist of unassembled short reads (~300bp). Thus, a tBLASTn search was conducted to quantify the abundance of *gms* gene homologs in environment. The tBLASTn search parameters used in this study (e-value < $1e^{-20}$, identity > 60%, DNA sequence length > 80 bp) have been successfully applied by other studies (Saunders et al. 2019; Lee et al., 2016; Ogilvie et al., 2013.).

References:

1. Saunders, Jaclyn K., et al. Complete arsenic-based respiratory cycle in the marine microbial communities of pelagic oxygen-deficient zones. *Proceeding of the National Academy of Sciences of the United States of America* 14;116(2019):9925-9930.
2. Lee, Jin-Woo, et al. Metagenomic analysis reveals the contribution of anaerobic methanotroph-1b in the oxidation of methane at the Ulleung Basin, East Sea of Korea. *Journal of Microbiology* 54(2016):814-822.
3. Ogilvie, Lesley A., et al. Genome signature-based dissection of human gut metagenomes to extract subliminal viral sequences. *Nature Communications* 4(2013):2420.

5) The environmental dataset on GMGT-0 presented are not convincing. Firstly, the relative abundance of GMGT-0 (it is actually not clear if this is the only GMGT taken into account or if this includes GMGTs with cyclopentane moieties) is extremely low. In the SPM of the Baltic Sea and Black Sea it is never >0.2% of the total GDGTs (I guess they mean total isoprenoid GDGTs). This is extremely low and, in fact, generates analytical challenges to assess their relative amount. The profile of the Black Sea also shows the presence in oxic waters and a decrease in the euxinic waters. Actually, the Black Sea is 2200 m deep and the sampling stops at 110 m, so this sample set cannot reveal what is going on in this euxinic basin since only 5% of the depth range is covered. The SPM profile of the Baltic Sea contains only four datapoints and the two in the sub-oxic zone shows the highest relative abundance, whilst there is only one sample in the oxic zone obtained from the surface where the abundance of archaeal GDGTs is typically below detection limit. Hence, both SPM profiles do not contribute to the story. The data from modern wetlands shows indeed a higher relative abundance (ca. 4%) of GMGT-0 in anoxic soil but this dataset has been published in 2018 by Naafs et al., so this is telling nothing new ("we suggest that H-GDGTs are preferentially produced at depth, in the anoxic, water-saturated catotelm by anaerobic archaea and bacteria. This is consistent with the observation that all reported organisms that produce H-isoGDGTs are strict anaerobes"; quote from Naafs et al.). The data presented also only provide the relative abundance of

GMGT-0 whereas absolute concentrations (normalized on TOC) would be required; changes in the absolute abundance of GDGTs now also affect the relative abundance of GMGT-0. In conclusion, the environmental datasets do not contribute anything convincing or new and should be left out.

We understand reviewer's concerns about the complexity of developing a new biomarker, and agree to remove the GMGT lipid data, focusing on biology aspects.

Our initial proposal for developing GMGT as an anoxia proxy follows a logical sequence:

- 1) Discovery of GMGT Synthase: the identification of GMGT synthase enabled a comprehensive investigation of its biological sources by examining all archaeal genomes in the database, rather than relying on a few reported culture species.
- 2) *gms* Gene Present in Anoxic Environments: the presence of the GMGT synthase gene was exclusively detected in metagenomes from anoxic environments, suggesting a potential association between GMGT and anoxic conditions.
- 3) Enrichment of GMGT Lipids in Anoxic Environments: lipid analysis revealed an enrichment of GMGT lipids in anoxic environments, providing evidence of its potential as an anoxia biomarker.

This sequential approach demonstrates a systematic progression from gene discovery to environmental distribution, resulting in the recognition of GMGT as a promising indicator of anoxic conditions.

Following stage 2), our hypothesis suggested that GMGT lipids might exhibit a similar distribution pattern in anaerobic environments. To test this hypothesis, we had conducted an extensive search for published GMGT lipid data and reached out to collaborators who might possess relevant samples. Despite months of effort, it turns out that the quality of existing environmental GMGT lipid data did not meet the criteria set by reviewer for establishing GMGT lipid as an anoxia biomarker and proxy.

Our strategy to develop a new biomarker begins with the discovery of a novel lipid biosynthesis gene. This approach contrasts with the traditional procedure, which typically starts by correlating lipid profiles with specific environmental factors, and then explores their biological basis afterward. While our approach stands on a robust biological foundation from the outset, its weakness lies in the potential for low abundance of the target lipid biomarker in certain sites, leading to limitations in broader applicability. Nonetheless, by establishing the biomarker on a solid biological footing from the start, we aim to promote its reliability and effectiveness through extensive examination of environmental samples in follow-up studies.

In summary, we maintain confidence in the association between the distribution of GMGT lipids and environmental anoxic conditions, supported by their origins from anaerobic archaea and the distribution pattern of their *gms* gene in anoxic environments. However, we also realize the complexity of lipid distribution in nature, particularly cross geological time spans. Thus, we have agreed with the reviewer to remove current environmental GMGT lipid data from this study, recognizing the potential challenges and uncertainties associated with interpreting such data.

- 6) The whole story is framed as the data would provide a novel biomarker "for studying ocean

anoxic events dating back hundreds of millions of years in Earth's history" (see also the start of the introduction). I don't see the advantage of using GMGT-0 in the study of OAEs. We already have many biomarkers to indicate that the water column has been anoxic (e.g. dinorhopane, lycopene etc.) and some of these even allow us to assess that the lower part of photic zone was euxinic (e.g., isorenieratene derivatives; Sinninghe Damste and Koster, 1998; Kuypers et al., 2002). What is GMGT-0 offering in addition, assuming we are convinced it can only originate from anaerobic archaea? The authors already acknowledge themselves that they can't discriminate between archaea residing in the water column or in the sediment (that ultimately will become anoxic upon burial). So, what is going to reveal this in addition to what all the biomarker data on OAEs that have been obtained in the last 4-5 decades? I don't see the point and it is also not provided in the manuscript. If we look to the data of OAE2 (Demerara Rise), we again see that GMGT-0 (perhaps including GMGTs with cyclopentane moieties; no information provided) is again present in very low relative abundance (ca. 0.5% on average). This strongly contrasts the statement in the abstract "remarkably abundant in ancient sediments during past periods of oxygen deficiency, such as OAE-2". A slight increase is seen during OAE-2 but at this site bottom waters were probably also anoxic before and after the OAE, Otherwise, it would have been difficult to explain TOC levels of 10% throughout this section. This is also concluded on the basis of more established biomarkers in a recent paper (Abraham et al., 2023), which is co-authored by one of the co-authors of this manuscript. So, it remains totally unclear what the GMGT-0 profile is indicating.

Please review our above response. Moreover, even there are some other lipid biomarkers available for indicating environmental redox, we believe GMGT retains unique value as an anoxic proxy. Because our Gms phylogenetic analysis largely expanded the known range of GMGT producers (Fig.3a). Previously associated only with a few anaerobic thermophilic archaea, our research now identifies diverse archaeal phyla genetic capable of producing GMGT lipids. This broadens the potential presence of GMGT in various anoxic ecosystems and environments, unlike some established anoxic/euxinic biomarker, for instance, isorenieratene, which is limited to the photic zone. This suggests GMGT has the potential to be a more versatile tool to study environmental oxygen deficiency globally.

7) The scientific contribution of some of the authors also raises questions. The author contribution mentions "X.L., H.Y., J.K., G.T.C., A.K.W., B.D.A.N., L.D. and F.W. provided environmental samples and performed lipid analysis" but as far as I can see there is only data of three co-authors: JK and AKW (environmental GDGT and GMGT data Black Sea and Baltic Sea; Fig. 4a) and BDAN (GDGT and GMGT data Modern Wetland (Fig. 4b) and OAE2 Demerara Rise (Fig. 4c). The GDGT data for these sites have previously been published in Wittenborn et al (2023), Naafs et al. (2018), and Abraham et al. (2023). The relative quantitation of GMGT-0 in these cases only required the integration of one additional peak in their dataset (in case of Naafs et al., 2018 this was already reported). So, it remains unclear why there is such an extensive experimental description of GDGT/GMGT analysis in this manuscript; the authors could have just referred to these papers. The contribution of the other five authors remains a mystery. There is a reference to a paper of Connock et al. (ref 45) in the legend of Fig. 4 but this paper refers to a site in the South Atlantic and not in the proto North Atlantic Ocean where Demerara Rise is located.

The author list has been changed to accompany with the removal of environmental GMGT lipid data.

I would strongly suggest focusing on the strong point of the paper (the biochemical characterization of the enzyme and apparent limitation to anaerobic archaea) and leave the rest out.

As explain above, we have removed all the environmental GMGT lipid data, and focused on the rest aspects.

More detailed comments:

Line 1 Title doesn't read well. Furthermore, the authors do not provide any mechanism why GMGT-0 is only produced in anaerobic archaea. The title should reflect what has really been demonstrated (e.g. "Anaerobic archaea produce GMGT-0 by a novel biphytanyl crosslinking enzyme").

The title has been changed to "Biosynthesis of GMGTs by a radical SAM enzyme associated with anaerobic archaea and environmental oxygen deficiency".

Line 30 "Often overlooked subset" Why?

We mean the physiology and geological significance of GMGT are not clear, and thus researchers pay less attention to GMGT compared to cyclic GDGTs. We have removed these words to prevent confusion.

Line 31 Clumsy writing: it is the enzyme that is primarily responsible. This sentence can be written in a more clear way.

Edits made as suggested.

Lines 41-44 Very speculative and not supported by the data provided.

These statements have been removed.

Lines 47-70 The introduction has no focus and is poorly written (e.g. no connection between paragraph #1 and #2).

The induction was rewritten, the paragraph 1# (oxygen deficiency content) has been removed.

Line 55 If you want to refer to a paper that applies GDGT for SST reconstruction during OAEs an original paper needs to be quoted. Ref 4 does not make sense in this respect.

OAEs content has been removed, and the introduction has been rewritten, and Schouten et al., OG, 2013 review paper has been cited.

Line 66 "extra cross link"? How do you mean? GDGDT have no cross-link

We tend to express GMGT has an extra cross link on isoprenoid than GDGT.

The sentence has been revised to "distinguished by an additional C-C linkage between the two isoprenoid chains".

Line 69 “synthetic mechanisms”. Unclear what this means.

It has been revised to “biochemical mechanisms”

I have only annotated page 1-2. This paper has 20 authors, some are based on highly ranked universities where English is the mother tongue. I can hardly believe that all of them have thoroughly gone through the text and made scientific and editorial comments. It is not the task of a referee to do this job.

We have done careful proofreading and corrected language errors.

Jaap Sinninghe Damsté

February 5, 2024

Abraham, M. A. F., Naafs, B. D. A., Lauretano, V., Sgouridis, F., & Pancost, R. D. (2023). Warming drove the Expansion of Marine Anoxia in the Equatorial Atlantic during the Cenomanian Leading up to Oceanic Anoxic Event 2. *Clim. Past*, 19, 2569–2580.

Kuypers, M. M., Pancost, R. D., Nijenhuis, I. A., & Sinninghe Damsté, J. S. (2002). Enhanced productivity led to increased organic carbon burial in the euxinic North Atlantic basin during the late Cenomanian oceanic anoxic event. *Paleoceanography*, 17(4), 3-1.

Lutnaes, B. F., Krane, J., Smith, B. E., & Rowland, S. J. (2007). Structure elucidation of C 80, C 81 and C 82 isoprenoid tetraacids responsible for naphthenate deposition in crude oil production. *Organic & Biomolecular Chemistry*, 5(12), 1873-1877.

Naafs, B. D. A., McCormick, D., Inglis, G. N., & Pancost, R. D. (2018). Archaeal and bacterial H-GDGTs are abundant in peat and their relative abundance is positively correlated with temperature. *Geochimica et Cosmochimica Acta*, 227, 156-170.

Sinninghe Damsté, J. S., & Köster, J. (1998). A euxinic southern North Atlantic Ocean during the Cenomanian/Turonian oceanic anoxic event. *Earth and Planetary Science Letters*, 158(3-4), 165-173.

Wittenborn, A. K., Bauersachs, T., Hassenrück, C., Käding, K., Wäge-Recchioni, J., Jürgens, K., ... & Kaiser, J. (2023). Nitrosopumilus as main source of isoprenoid glycerol dialkyl glycerol tetraether lipids in the central Baltic Sea. *Frontiers in Microbiology*, 14.

Reviewer #2 (Remarks to the Author):

This manuscript is about the identification of the enzyme responsible of converting the membrane-spanning tetra ether lipid GDGT-0 into the bridged derivative H-GDGT-0. Through heterologous expression and in vivo lipid analysis, as well as protein purification and in vitro enzyme assays, the authors convincingly show that this C-C bonding enzyme belongs to the radical SAM class of enzymes, elucidating a further enzymatic step in archaeal lipid biosynthesis.

comments

Fig. 1b, the in vitro activity of the purified enzyme seems very low. Likely the enzyme readily inactivates when exposed to oxygen; or cofactor incorporation might be very poor. The authors should consider the optimised conditions for radical SAM protein expression as described in ref 6. This is important as now it is not entirely clear if the further analysis (EPR) was done on a fully functional enzyme.

Otherwise, in this in vitro assay, it could be an issue of substrate exposure to the enzyme. I noticed the assays do not contain detergent which would be necessary to keep the GDGT-0 monodispersed.

We performed the in vitro activity assay under anaerobic conditions. And actually, both expression and purification procedures were carried out in the anaerobic chamber to avoid any damage due to oxygen. We have specifically emphasized this point now.

The EPR spectrum clearly shows signals from all three FeS clusters, suggesting that the enzyme we purified is fully active.

For the in vitro activity assay, we first dissolved GDGT-0 in DMSO, and then slowly added it to the enzyme in a buffer solution. For complete incubation of GDGT-0 within the enzyme, we monitored the mixture to make sure it reached a homogeneous state before adding SAM to initiate the reaction. Although the in vitro activity is lower than in vivo activity, this could be due to the slow-release process of the product.

The authors propose a putative mechanism using the alfaFold2 model and insights on other radical SAM proteins. However, what I entirely miss in the discussion is how this enzyme can act on a substrate that supposedly is membrane spanning. How does the binding site even get access to the relevant part of the substrate, that in the membrane spanning state will be in the middle of the membrane! A transparent discussion is needed.

By comparing the structure of Gms to Tes, SuiB, and GrsA, we found some structural uniqueness of Gms and speculated that its β -hairpin might be involved in anchoring the protein to the membrane. A detailed discussion has been added to the revised manuscript (see new Supplementary Fig. 8, and line: 149-170).

Reviewer #3 (Remarks to the Author):

Glycerol dialkyl glycerol tetraethers (GDGTs) are a special class of archaeal ether lipids that are membrane-spanning and can have several modifications/variations like crosslinks between the isoprenoid chains (H-GDGTs) or rings within the isoprenoid chains. Only in the last few years, the discovery of the enzymes that catalyze the reactions at the isoprenoid chains began (Tes, GrsA, GrsB), they are all radical SAM (RS) enzymes. The presented manuscript is a continuation of the elucidation of these interesting enzymes, as it describes for the first time the RS enzyme that introduces the crosslinks in H-GDGTs, why the authors call it "H-HDGT bridge synthase" (Hbs).

Given Reviewer 1 raised similar concern about the terminology, we have used “GMGT” , instead of “H-GDGT”, as the primary term, and renamed the enzyme as “GMGT synthase (Gms)”.

The authors study the Hbs activity in vitro and in vivo. Furthermore, they explore the occurrence of Hbs homologues in fully sequenced species and in metagenomic data. They also provide new own data on the occurrence of H-GDGTs in oxygen-deficient and oxic water samples. From all those data, they deduce that H-GDGT producing species are obligate anaerobic, and therefore only occur in (almost) oxygen-free environments. They suggest that H-GDGTs might be suitable as a new biomarker for studying oxygen deficiency in the geological past.

The study is, in most parts, properly done, and the results are properly presented. However, I have one main concern. Minor issues are listed below.

Main concern:

The authors claim that Hbs homologues only occur in obligate anaerobic species (l. 171 ff), which is stressed as one of the major conclusions from this study. For this, they do BLAST searches within archaeal genomes with defined search parameters (e-value limit, min. identity, ...). The authors do, however, not prove that these parameters are suited to distinguish between “true” Hbs and other related RS enzymes like for example Tes that also occur in aerobic species. The authors must somehow show that the hits they get are homogenous in their sequence properties, but different from Tes, Grs, or other, not yet characterized RS enzymes. This could, for example, be done by using the sequence similarity network (SSN) technique, or at least by showing adequate multiple sequence alignments or phylogenetic trees with all these RS enzymes (not only Grs as an outgroup as in ext. data Fig. 7). The same applies for the search in the metagenomes. For clarity, the authors should not only show the hits among metagenomic data in source data (is this really what is shown according to l. 606 in “Archaeal domain-Hbs-homologs”?) but also all the hits from the BLASTP search.

Following these suggestions, we have used the SSN method to distinguish Gms from Grs and Tes. The SSN analysis results clearly indicated that Gms form distinct clusters separate from Grs or Tes enzymes. Additionally, sequence comparison revealed minimal homology between Gms and Grs or Tes. As a result, it is unlikely that Gms homology searches would include any Grs or Tes enzymes. For detailed information, please refer to the new supplementary Fig. 10 and included below.

Supplementary Fig. 10 | The homology analysis of Gms, Tes, and Grs sequences.

In our search for similarity with other radical SAM proteins, we found that Gms sequence shares similarity with AhbC and AhbD proteins (32–39% identities). These two proteins are

known to be involved in the anaerobic biosynthesis of heme b in methanogens (anaerobic archaea) and sulfate-reducing bacteria (anaerobic bacteria). However, Gms protein possesses an additional ~100 amino acid sequence in the N-terminal region compared to AhbC and AhbD. As a result, Gms can be distinguished from AhbC and AhbD by setting a length cutoff of 420aa in the search parameters. Furthermore, in the SSN method, using an alignment score of 90 is sufficient to differentiate Gms from AhbC and AhbD. For detailed information, please refer to the new supplementary Fig. 11 and included below.

Supplementary Fig. 11 | The homology analysis of Gms and AhbC/AhbD.

In source data, “Archaeal domain-Gms-homologs” is the hits of BLASTP from archaeal genome database, and the “Ocean-Gms (RPKM)” and “Lake-Gms (RPKM)” are the hits of metagenomes search with TBLASTN.

In general (see I.130ff, and also 1st paragraph of discussion), the whole story would benefit from a more detailed description of the structural properties of Hbs in comparison with Tes

(which also has auxiliary Fe-S clusters), GrsA/B (which have no auxiliary Fe-S clusters) and SuiB. This could be done as extensions to Fig. 1 and ext. data Fig. 1, the EPR data then could be a separate figure.

We have incorporated a more detailed description of the structural comparison with Tes, SuiB, and GrsA, along with the addition of a new supplementary Fig. 8.

For the suggestion of separation of protein structure and EPR data into two figures, we believe that displaying the EPR data together with the structure model in one figure would make it easier for readers to understand the signals of each cluster. EPR is a unique spectroscopic technique used for characterizing paramagnetic FeS clusters, and presenting the data alongside the structure model would provide a better context for its interpretation. Therefore, we would like to proceed with presenting the EPR data with the structure model in one figure.

Other issues, in order of appearance:

Carefully read again the manuscript for good language and typos, especially ll. 530-545 are full of them.

Errors have been corrected.

ll. 78-81: The result of this comparison should be better presented

We have added a new supplementary table 4 to list this comparison.

l. 82f: a gene cannot contain a motif defined by amino acids, only a protein can contain such a motif

Edits made as suggested.

l. 121 ff: As EPR is a rarely seen method, it might be helpful for readers not so familiar with it to very briefly resume at the beginning of the paragraph what can be measured with this method.

We have added a brief introduction to EPR before presenting the EPR results. "EPR spectroscopy is a technique that only probes paramagnetic species, e.g., organic radical, metal ions with specific oxidation states, or multi-nuclear cluster in certain redox states."

ext. data Fig. 1: At the time of the first reference to this figure in the text, nobody knows yet that WP_62369926.1 is Hbs, so both names should be used in the figure legend. It must also be introduced in the legend that the structure is a model, as also this is only introduced later in the text.

Edits made as suggested.

Fig. 2: The language of the legend should be improved (e.g.: "a, Four domains shown in Hbs").

The legend language has been revised.

l. 172/173: a) BLASTP can only search in protein databases, but not in "archaeal genomes" = DNA sequences; b) Is it really the case that the search was limited to archaea? This does not

become clear from the according methods section. What about hits in bacteria? This might be interesting for some people because Tes homologues also occur in bacteria.

Indeed, we conducted a search for Gms protein homologs in the NCBI "non-redundant protein sequence" database. The NCBI BLASTP website enables searches within specific organism genomes, as illustrated in the figure below. For this study, we restricted the search to archaeal genomes, resulting in Gms homologs exclusively from archaea. We have revised the method of this part to avoid confusion.

While we did find Gms homologs are present in some bacterial genomes. However, bacteria do not synthesize archaeal-type GDGT or GMGT. Therefore, these bacterial Gms homologs likely have different biological function with different substrate. Studying the function of these bacterial Gms homologs falls outside the scope of this study.

l. 174: the authors state that they got 649 hits. In methods, they state that they cluster sequences with 90% similarity, in ext. data Fig. 7 they mention 177 unique homologues from 910 genomes. How do those numbers fit together, especially the 910 genomes?

The 177 Gms homolog hits used for the construction of phylogenetic tree (previous ext. data Fig. 7) were obtained from JGI IMG database through BLASTP searches conducted on 910 archaeal genomes. However, the 649 hits used for Fig. 3A were acquired from NCBI database. It's not unexpected that conducting BLASTP searches for Gms in two different databases yielded different numbers of hits (see below figure).

To be consistent, we have constructed a new phylogenetic tree with Gms homologs from NCBI database BLASTP search (see new Supplementary Fig. 9).

ext. data Fig. 7: "Based on 177 unique archaea Hbs homologues and 19 archaea Gms

homologues from 910 archaea genomes in JGI IMG 691 database as outgroups [...]”: the sentence as a whole seems strange: are the 177 unique homologues from the 910 genomes, or the 19 Grs sequences, or both together? Why more genomes than sequences?

The legend of supplementary Fig. 9 has been revised to clarify confusion (Supplementary file line: 85-91). Because only a subset of archaeal genomes contains the Gms gene or protein, resulting in fewer total Gms hits than the total number of archaeal genomes.

ext. data Fig. 7: I cannot see any blue circles as mentioned in the legend.

A new Gms phylogenetic tree has been constructed as described above with improved figure resolution (see new Supplementary Fig. 9).

ext. data Fig. 7: It would be appropriate to list also the Grs sequences in “source data”. With respect to my main concern: Would Tes as an outgroup also work?

In our response to your above main concern, we have compared the similarity between Gms with other RS proteins including Tes and Grs. Here, we constructed a Gms phylogenetic tree using Tes sequences as outgroup (below figure). These results showed both Grs and Tes sequences work fine as an outgroup in Gms phylogenetic tree construction.

l. 212: ext. data Fig. 8b? 8c does not exist.

Thanks for pointing this out, and we have corrected that.

Fig. 3b, c: What is the y-axis of these diagrams? Water/Sediment depth like in Fig. 4?

Yes, and we have added instruction in figure legend: "The y-axis represents the approximate depths of water and sediment." (line 252-253)

l. 249ff: Shouldn't equation 1 better already appear two paragraphs above (ll. 236-241)? Or move it to methods?

We have removed these environmental lipid data in response to the Reviewer 1's request.

l. 308f: "The presence of H-GDGTs primarily depends on shift in its associated microbial communities in the water column or sediment, which are highly sensitive and responsive to the rise of anoxic condition.": I do not completely understand the meaning of this sentence. Better: ... depends on the distribution of the species that produce the H-GDGTs?

Thanks for the suggestion, but this paragraph discussing environmental lipid data has been removed in response to the Reviewer 1's request.

l. 312 f.: "[...] the H-GDGTs proxy has the advantage to yield a more precise and regional representation of ocean oxygen deficiency supported by its well-established biological basis." I also do not understand this completely. What is meant with "its well-established biological basis"?

Like above, this paragraph has been removed.

l. 539 and elsewhere: "centrifugation at 6,000 rpm for 30 min at 4 °C": it makes only sense to give the rpm if you also mention the rotor. Alternatively, give the g value.

The speed has been converted to g value in manuscript.

l. 593: is really clustered at 90% sequence similarity (using a scoring matrix) or rather at 90% sequence identity?

We have revised that to "90% sequence identity".

Ext. data Table 1: separating horizontal lines between the M. maripaludis, E. coli and S. acidocaldarius strains might improve clarity. Typo "Escherichia coil".

Edits made as suggested.

Reviewer #1 (Remarks to the Author):

See attached pdf

Reviewer #1 Attachment on the following page

Second review of Li et al. “Biosynthesis of GMGTs by a radical SAM enzyme associated with anaerobic archaea and environmental oxygen deficiency”.

The authors have adjusted the manuscript extensively according to the comments of the three referees. However, there remain a number of issues that prevent me to recommend this version for publication. Firstly, I will provide a response to the rebuttal of my previous five comments.

- 1) It is good to see that the authors have taken this comment seriously and changed the name of the gene and enzyme accordingly. Nevertheless, they still use “H-branched” in a wrong way (e.g., line 50 “The H-shaped GDGTs”). They should only apply this terminology to the lipid without the two glycerol units (i.e. as identified by Lutnaes et al, 2006, 2007). There the structure forms a real H.
- 2) I remain with my criticism that it is strange to establish the enzyme producing a biochemical product that has not been fully characterized chemically. This also complicates the discussion because one cannot be sure if indeed, as assumed, the enzyme catalyzes the reaction bridging two biphytanyl chains by linking two methyl groups through a covalent C-C bond. The authors have isolated GDGT-0 from an archaeal culture to serve as substrate for their in vitro experiment (Fig. 1b). Their *M. maripaludis* plasmid-*gms-tes* experiment shows a higher abundance of GMGT-0 than GDGT-0, so why can't they isolate GMGT-0 and prove its structure by 2D NMR? That would really justify the publication of this manuscript in Nature Comm.
- 3) The authors have now adequately described how they arrive at the conclusion that GMGT-0 is “only” produced by anaerobic archaea although a clear explanation of why this is the case is still lacking. Another issue is that it remains unclear whether facultatively anaerobic archaea are capable of producing GMGT-0. Figure 3A shows that some are indeed potentially capable of producing GMGT-0, whereas on basis of published lipid studies they conclude “the biosynthesis of GMGTs is exclusive to **obligate** anaerobic archaea” (lines 207-208). So, this remains completely unclear. One would expect that the authors would have grown some facultatively anaerobic archaea under different conditions to see what the effect of oxygen concentration on the production of GMGT-0 (i.e., the degree of crosslinking of GDGT-0) is. Only with such experiments in hand a valid discussion on this topic can be made.
- 4) The authors have adequately explained this. The figure on the environmental abundance of the *gms* gene (Figure 3b), however, raises again the question with respect to potential production of GMGT-0 in sub-oxic environments (i.e. by archaea that can withstand low levels of oxygen). See point 3. Also, the absence of a depth scale for the water column in Fig. 3b is not acceptable.
- 5) The authors have followed my suggestion.
- 6) Even though the emphasis on OAEs has been removed, the authors remain in their conclusion that GMGT-0 can be used as an indicator for “environmental deoxygenation” (lines 282-289). See also last sentence (“demonstrate the potential of GMGTs to be a biomarker for indicating oxygen-deficient environments”.) Assuming that GMGT-0 is indeed only produced by anaerobic archaea, the presence of GMGT in an ancient sediment does not tell a lot since it

can be produced by archaea residing in the suboxic water column, the anoxic water column, the anoxic sediment-water interface **or** deeper in the anoxic part of the sediment. All sediments upon burial become ultimately anoxic and the deep biosphere contains various groups of archaea that have the *gms* gene. So, it remains totally unclear what we learn from this. Consequently, all speculation about GMGT-0 as environmental indicator should be left out. It just weakens the manuscript. I would also suggest to take out “and environmental oxygen deficiency” from the title because this also hints to the use of GMGT-0 as environmental indicator. In addition, the fact that it is “associated **anaerobic** archaea” already implicitly indicates that it will occur in oxygen deficient environments. I still feel that the title I suggested (“Anaerobic archaea produce GMGT-0 by a novel biphytanyl crosslinking enzyme”) provides a better summary of the content of the work.

- 7) Author list has changed substantially according to my comments. There are still two American co-authors from Stanford University who have contributed to the writing (line 593). Nevertheless, the English needs to be improved substantially.

Other comments.

Line 43 It is not correct that cyclization of GDGTs catalyzed by the enzyme encoded by *GrsA/B* is a C(sp³)-C(sp³) cross-linkage reaction as stated. This is a C(sp³)-C(sp²) cross-linkage reaction. This also has implication for later discussion in the manuscript (line 265-267).

Line 60 Not every compound has a “potential significance as a reliable environmental proxy”.

Line 85 An important piece of information that is missing here is: does *Methanothermococcus okinawensis* produce GMGT-0? This should be mentioned based on the literature or tested.

Line 90 Clumsy language

Line 188 “To identify the biological sources of GMGTs”? Rephrase, a biological source has already been identified.

Line 207 Actually, this is a quite limited dataset with respect to production of archaeal GMGTs.

Lines 212 Leave out “ecosystems”.

Lines 227-8 This is further reducing the potential of GMGT-0 as biomarker for “anoxic conditions”.

Line 257 You cannot say this because the structure of GMGT-0 remains unproven (see before).

May 3, 2024

Jaap S. Sinninghe Damste

Reviewer #2 (Remarks to the Author):

[No further comments for authors]

Reviewer #3 (Remarks to the Author):

The authors have made improvements to the manuscript. However, I am not fully satisfied yet with their responses to my main concerns. In general, the authors should bring their arguments in the manuscript, not only in the rebuttal letter, so that all readers can follow.

The authors now use SSNs from RadicalSAM.org to classify different RS enzymes (suppl. Figs. 10 and 11), which is a step in the right direction. However, they now mix up two different datasets in their argumentation whether Gms homologues only occur in anaerobic archaea. For this conclusion, the authors still rely on the data shown in Fig. 3a and suppl. Fig. 9, which is based, unchanged to in the initial version of the manuscript, on a BLAST search in the nr database. For this sequence set, it is still not clear whether it contains only "true" Gms sequences. The authors only argue in their rebuttal letter, but not in the manuscript, that the parameters used in the BLAST search might be sufficient to exclude other sequences than "true" Gms, and I can't completely follow this argumentation yet. If I understand everything correctly, only the sequences from the SSN cluster "Mega 1-1-24" (suppl. Fig. 11) are "true" Gms sequences. The authors either should use this sequence set for their argumentation, or generate a new SSN from the sequences used for Fig 3a/suppl. Fig 9, and correlate it with the SSNs provided by RadicalSAM.org. This would be a proper way to demonstrate that the sequences form a unique cluster.

The same applies for the metagenome analysis.

other issues concerning the SSN part:

Methods part, l. 426 ff, Fig 10A, Fig 11A, corresponding main text: how were the SSNs generated, or better, how did the authors of ref. 20 generate it? How are the enzymatic functions (Gms, Tes, ...) assigned to the clusters? How is the statement "Gms [...] can be separated from other identified radical SAM enzymes" deduced from the data? I think somebody not familiar with SSNs cannot follow the argumentation.

Suppl. Fig. 10 B, C, 11B: "Grs vs. Gms 20.98%" and similar. Is this sequence similarity or identity? What algorithm was used to align the sequences?

Suppl. Fig. 11b: according to the legend, this has been created using "NCBI BLASTP search using EFI-EST". How and what settings have been used? Why do you only get Gms and AhbC/AhbD here? AhbC/AhbD are only mentioned by their name in the methods section and in Fig S11B, but without any explanation. What can the reader learn from this figure?

new suppl. table 4:

I can only assume (because there is no explanation in the manuscript) what can be learned from this table. The table is not mentioned anywhere in the text. I asked for a better presentation of the comparison mentioned in l. 72ff, because I could not follow the argumentation (because there is no argumentation in the text), why this comparison led to the identification of WP_062369926.1 as an interesting candidate. I think this misunderstanding starts with a misleading wording. The authors write: "We compared radical SAM enzymes (pfam04055) between two archaeal groups: one group involving *Pyrococcus furiosus* and *Thermococcus guaymasensis* produces GMGTs, while the other group including *Pyrococcus yayanosii* and *Thermococcus kodakarensis* does not." According to suppl. Table 4, the authors did not compare "archaeal groups" what I first understood as "groups of different species", where the one produces GMGT, the other not, but (maybe?) they compared all (?) RS enzymes that occur in the four archaeal species listed in the table. Are the two "hot candidates" marked in the table simply the sequences where no closest homologues occur in *P. yayanosii* and *T. kodakarensis*? Or what else makes them interesting?

additional issues:

Please again check for correct spelling and clear language.

e.g.

Legend to Fig. 1B: "retention time", not "retension"

for clear language see as an example the sentence I cited in the paragraph on suppl. table 4.

The sequences of the outgroup sequences for suppl. Fig. 9 are still not included in the source data file.

Second review of Li et al. "Biosynthesis of GMGTs by a radical SAM enzyme associated with anaerobic archaea and environmental oxygen deficiency".

The authors have adjusted the manuscript extensively according to the comments of the three referees. However, there remain a number of issues that prevent me to recommend this version for publication. Firstly, I will provide a response to the rebuttal of my previous five comments.

1) It is good to see that the authors have taken this comment seriously and changed the name of the gene and enzyme accordingly. Nevertheless, they still use "H-branched" in a wrong way (e.g., line 50 "The H-shaped GDGTs"). They should only apply this terminology to the lipid without the two glycerol units (i.e. as identified by Lutnaes et al, 2006, 2007). There the structure forms a real H.

All "H-GDGTs" terms have been removed from new manuscript.

2) I remain with my criticism that it is strange to establish the enzyme producing a biochemical product that has not been fully characterized chemically. This also complicates the discussion because one cannot be sure if indeed, as assumed, the enzyme catalyzes the reaction bridging two biphytanyl chains by linking two methyl groups through a covalent C-C bond. The authors have isolated GDGT-0 from an archaeal culture to serve as substrate for their *in vitro* experiment (Fig. 1b). Their *M. maripaludis* plasmid-gms-tes experiment shows a higher abundance of GMGT-0 than GDGT-0, so why can't they isolate GMGT-0 and prove its structure by 2D NMR? That would really justify the publication of this manuscript in Nature Comm.

In the *in vitro* experiment, the substrate GDGT-0 was isolated from the culture of a *Sulfolobus* mutant strain, which produces large amount of GDGT-0 as its major membrane lipid. However, only a small proportion of GDGT-0 was converted to GMGT-0. On the other hand, the *Methanogen* heterologous express only produced a small amount of GDGT-0, although a decent proportion of GDGT-0 was converted to GMGT-0. Therefore, neither the *in vitro* nor the *in vivo* experiments could generate a sufficient amount of GMGT-0 for NMR analysis.

3) The authors have now adequately described how they arrive at the conclusion that GMGT-0 is "only" produced by anaerobic archaea although a clear explanation of why this is the case is still lacking. Another issue is that it remains unclear whether facultatively anaerobic archaea are capable of producing GMGT-0. Figure 3A shows that some are indeed potentially capable of producing GMGT-0, whereas on basis of published lipid studies they conclude "the biosynthesis of GMGTs is exclusive to obligate anaerobic archaea" (lines 207-208). So, this remains completely unclear. One would expect that the authors would have grown some facultatively anaerobic archaea under different conditions to see what the effect of oxygen concentration on the

production of GMGT-0 (i.e., the degree of crosslinking of GDGT-0) is. Only with such experiments in hand a valid discussion on this topic can be made.

Based on published studies regarding the respiration features in all archaea containing Gms homolog, none of these archaea were described as facultative organisms (see Fig. 3a, blue circle: Aerobic/Facultative). Furthermore, *Sulfolobus* species, which are representative facultative archaea, do not contain Gms and do not produce any GMGTs.

4) The authors have adequately explained this. The figure on the environmental abundance of the gms gene (Figure 3b), however, raises again the question with respect to potential production of GMGT-0 in sub-oxic environments (i.e. by archaea that can withstand low levels of oxygen). See point 3. Also, the absence of a depth scale for the water column in Fig. 3b is not acceptable.

Many studies have showed the presence of obligate anaerobic microbes in suboxic environments (e.g., OMZ), and have suggested anaerobic microbes can tolerate suboxic conditions periodically (Thamdrup et al., 2019; Jensen et al., 2008). Therefore, the detection of the *gms* gene in suboxic environments does not contradict the conclusion that GMGTs producers are obligate anaerobic archaea.

To enhance the visualization and readability of Figure 3b, we aligned all water stratification using oxygen levels instead of using the actual depth parameter for all data sites. The depth information for each data point is provided in the Source Data Excel file.

References:

1. Thamdrup, Bo, et al. "Anaerobic methane oxidation is an important sink for methane in the ocean's largest oxygen minimum zone." *Limnology and Oceanography* 64.6 (2019): 2569-2585.
2. Jensen, Marlene M., et al. "Rates and regulation of anaerobic ammonium oxidation and denitrification in the Black Sea." *Limnology and Oceanography* 53.1 (2008): 23-36.

5) The authors have followed my suggestion.

6) Even though the emphasis on OAEs has been removed, the authors remain in their conclusion that GMGT-0 can be used as an indicator for "environmental deoxygenation" (lines 282-289). See also last sentence ("demonstrate the potential of GMGTs to be a biomarker for indicating oxygen-deficient environments".) Assuming that GMGT-0 is indeed only produced by anaerobic archaea, the presence of GMGT in an ancient sediment does not tell a lot since it can be produced by archaea residing in the suboxic water column, the anoxic water column, the anoxic sediment-water interface or deeper in the anoxic part of the sediment. All sediments upon burial become ultimately anoxic and the deep biosphere contains various groups of archaea that have the gms gene.

So, it remains totally unclear what we learn from this. Consequently, all speculation about GMGT-0 as environmental indicator should be left out. It just weakens the manuscript. I would also suggest to take out “and environmental oxygen deficiency” from the title because this also hints to the use of GMGT-0 as environmental indicator. In addition, the fact that it is “associated anaerobic archaea” already implicitly indicates that it will occur in oxygen deficient environments. I still feel that the title I suggested (“Anaerobic archaea produce GMGT-0 by a novel biphytanyl crosslinking enzyme”) provides a better summary of the content of the work.

We have removed the statement regarding GMGTs lipid's potential as an environmental oxygen deficiency biomarker/indicator from the manuscript. However, we insist on keeping the statement that “GMGTs producers are associated with oxygen-deficient habitats”, because this assertion is supported by both biological source analysis and metagenome analysis (Fig. 3).

7) Author list has changed substantially according to my comments. There are still two American co-authors from Stanford University who have contributed to the writing (line 593). Nevertheless, the English needs to be improved substantially.

We have further refined the writing.

Other comments.

Line 43 It is not correct that cyclization of GDGTs catalyzed by the enzyme encoded by GrsA/B is a C(sp³)-C(sp³) cross-linkage reaction as stated. This is a C(sp³)-C(sp²) cross-linkage reaction. This also has implication for later discussion in the manuscript (line 265-267).

We do not fully understand this comment, but we want to clarify that a linkage forming between a methyl group and an ethyl group is indeed a C(sp³)-C(sp³) linkage, not a C(sp³)-C(sp²) linkage.

Here are more biochemical details about the reaction GDGTs cyclization. Due to the lack of in vitro biochemical analysis of GDGT cyclization catalyzed by Grs proteins, it has not been fully identified whether their substrate GDGT-0 requires the presence of double bonds to facilitate cyclization for forming a C-C linkage ring. One hypothesis suggests that Grs proteins induce GDGT cyclization by forming a C(sp³)-C(sp²) linkage on unsaturated GDGTs (with double bonds) (Pearson, 2019). Alternatively, Grs proteins could induce GDGT cyclization by forming a C(sp³)-C(sp³) linkage on saturated GDGTs (without double bonds). Recent studies have shown that radical SAM proteins could generate C(sp³)-C(sp³) linkages during GDGTs biosynthesis, such as Tes, and Gms in this study. Thus, it is more reasonable to hypothesize that GDGT cyclization involves a C(sp³)-C(sp³) linkage reaction.

We have revised the language to clearly demonstrate that the C(sp³)-C(sp³) linkage reaction on GDGT cyclization is presented as a hypothesis rather than an assertion.

Reference:

1. Pearson, Ann. "Resolving a piece of the archaeal lipid puzzle." Proceedings of the National Academy of Sciences 116.45 (2019): 22423-22425.

Line 60 Not every compound has a “potential significance as a reliable environmental proxy”.

Since the GMGT lipid proxy content has been removed, this statement has been revised as “Nevertheless, the lack of knowledge regarding their biosynthesis, biochemical mechanisms, and comprehensive biological sources hinders our understanding of their physiological significance.”

Line 85 An important piece of information that is missing here is: does *Methanothermococcus okinawensis* produce GMGT-0? This should be mentioned based on the literature or tested.

Yes, this organism has been reported to produce GMGT-0 by Baumann et al., and we have mentioned and cited this paper in our manuscript (line 87).

Reference:

1. Baumann, Lydia MF, et al. "Intact polar lipid and core lipid inventory of the hydrothermal vent methanogens *Methanocaldococcus villosus* and *Methanothermococcus okinawensis*." Organic geochemistry 126 (2018): 33-42.

Line 90 Clumpy language

Language has been improved.

Line 188 “To identify the biological sources of GMGTs”? Rephrase, a biological source has already been identified.

Only a few archaeal species were previously reported to produce GMGT lipids through culture and lipid analysis. However, in this study, the use of BLASTP for Gms enables a comprehensive survey of biological sources of GMGTs from various environments, significantly expanding the diversity of organisms possessing the genetic capability to produce GMGTs.

Line 207 Actually, this is a quite limited dataset with respect to production of archaeal GMGTs.

This sentence has been revised to express suggestion more accurately.

Lines 212 Leave out “ecosystems”.

Edit has been made as suggested.

Lines 227-8 This is further reducing the potential of GMGT-0 as biomarker

for “anoxic conditions”.

The study of using GMGT lipid as anoxic proxy has been removed from this manuscript.

Line 257 You cannot say this because the structure of GMGT-0 remains unproven (see before).

Regardless of whether the C-C cross linkage occurs between two methyl groups, or between a methyl group and an ethyl group during GMGT synthesis, both scenarios involve a C(sp³)-C(sp³) linkage. Therefore, we believe that our statement regarding GMGT formation as the direct linkage of two sp³ carbons is appropriate.

May 3, 2024 Jaap S. Sinninghe Damste

Reviewer #3 (Remarks to the Author):

The authors have made improvements to the manuscript. However, I am not fully satisfied yet with their responses to my main concerns. In general, the authors should bring their arguments in the manuscript, not only in the rebuttal letter, so that all readers can follow.

We have added more descriptions about the use of SSNs and its results and methods in manuscript. Please see line 194-205, and line 438-441.

The authors now use SSNs from RadicalSAM.org to classify different RS enzymes (suppl. Figs. 10 and 11), which is a step in the right direction. However, they now mix up two different datasets in their argumentation whether Gms homologues only occur in anaerobic archaea. For this conclusion, the authors still rely on the data shown in Fig. 3a and suppl. Fig. 9, which is based, unchanged to in the initial version of the manuscript, on a BLAST search in the nr database. For this sequence set, it is still not clear whether it contains only “true” Gms sequences. The authors only argue in their rebuttal letter, but not in the manuscript, that the parameters used in the BLAST search might be sufficient to exclude other sequences than “true” Gms, and I can’t completely follow this argumentation yet. If I understand everything correctly, only the sequences from the SSN cluster “Mega 1-1-24” (suppl. Fig. 11) are “true” Gms sequences. The authors either should use this sequence set for their argumentation, or generate a new SSN from the sequences used for Fig 3a/suppl. Fig 9, and correlate it with the SSNs provided by RadicalSAM.org.

This would be a proper way to demonstrate that the sequences form a unique cluster.

Following this suggestion (“or generate a new SSNs from the sequences...”), we have re-analyzed and updated the Fig. 3a and Supplementary Fig. 9 (phylogenetic tree) by using SSNs cluster “Mega 1-1-24” dataset (413 Gms homologs).

The same applies for the metagenome analysis.

The SSNs analysis cannot be applied to environmental metagenome dataset in this study (SRA, Sequence Read Archive), because this dataset consists of unassembled short reads (~300bp).

other issues concerning the SSN part:

Methods part, l. 426 ff, Fig 10A, Fig 11A, corresponding main text: how were the SSNs generated, or better, how did the authors of ref. 20 generate it? How are the enzymatic functions (Gms, Tes, ...) assigned to the clusters? How is the statement “Gms [...] can be separated from other identified radical SAM enzymes” deduced from the data? I think somebody not familiar with SSNs cannot follow the argumentation.

To enhance clarity and readability, we have rewritten the SSNs results and included additional background information on other radical SAM enzymes, such as AhbC/AhbD. (see line 194-205).

Supp. Fig. 10 B, C, 11B: “Grs vs. Gms 20.98%” and similar. Is this sequence similarity or identity? What algorithm was used to align the sequences?

They are sequence identity, and the alignment method (Clustal Omega) has been added in figure legend.

Supp. Fig. 11b: according to the legend, this has been created using “NCBI BLASTP search using EFI-EST”. How and what settings have been used? Why do you only get Gms and AhbC/AhbD here? AhbC/AhbD are only mentioned by their name in the methods section and in Fig S11B, but without any explanation. What can the reader learn from this figure?

Following the reviewer's suggestion, a new SSNs analysis has been conducted (see above response), which effectively distinguishes the “true” Gms from other radical SAM proteins. Thus, the analysis of protein length distribution in Gms and AhbC/D (previous Supplementary Fig. 11b) is unnecessary and has been removed.

new suppl. table 4:

I can only assume (because there is no explanation in the manuscript) what can be learned from this table. The table is not mentioned anywhere in the text. I asked for a better presentation of the comparison mentioned in l. 72ff, because I could not follow the argumentation (because there is no argumentation in the

text), why this comparison led to the identification of WP_062369926.1 as an interesting candidate. I think this misunderstanding starts with a misleading wording. The authors write: “We compared radical SAM enzymes (pfam04055) between two archaeal groups: one group involving *Pyrococcus furiosus* and *Thermococcus guaymasensis* produces GMGTs, while the other group including *Pyrococcus yayanosii* and *Thermococcus kodakarensis* does not.” According to suppl. Table 4, the authors did not compare “archaeal groups” what I first understood as “groups of different species”, where the one produces GMGT, the other not, but (maybe?) they compared all (?) RS enzymes that occur in the four archaeal species listed in the table. Are the two “hot candidates” marked in the table simply the sequences where no closest homologues occur in *P. yayanosii* and *T. kodakarensis*? Or what else makes them interesting?

We have rewritten this part to better describe how we obtained candidate gene (see line 72-78).

“It has been observed that *Thermococcus guaymasensis* produces GMGTs¹² while its close sister species *Thermococcus kodakarensis* does not¹³. Similarly, in the case of *Pyrococcus furiosus* and *Pyrococcus yayanosii*, the former produces GMGTs¹⁰, whereas the latter does not¹³. By comparing the homology of all radical SAM enzymes (pfam04055) between species with or without GMGTs formation, we successfully found a promising candidate protein (WP_062369926.1 from *T. guaymasensis*, or AAL80771.1 from *P. furiosus*), which is present in GMGTs-forming species but absent in species lacking GMGTs formation (Supplementary Table 1).”

additional issues:

Please again check for correct spelling and clear language.

e.g.

Legend to Fig. 1B: “retention time”, not “retension”

for clear language see as an example the sentence I cited in the paragraph on suppl. table 4.

Typo has been corrected.

The sequences of the outgroup sequences for suppl. Fig. 9 are still not included in the source data file.

The outgroup sequences have been added to Source Data file in “Grs_outgroup sequences” sheet.